# Deciphering spatial genomic heterogeneity at a single cell resolution in multiple myeloma

Maximilian Merz [1,2,14✉], Almuth Maria Anni Merz[1,14], Jie Wang[3,14], Lei Wei [3,14], Qiang Hu [3], Nicholas Hutson[3], Cherie Rondeau[1], Kimberly Celotto[1], Ahmed Belal[4], Ronald Alberico[4], AnneMarie W. Block[5], Hemn Mohammadpour[6], Paul K. Wallace[7], Joseph Tario[7], Jesse Luce[8], Sean T. Glenn[8], Prashant Singh[8], Megan M. Herr [9], Theresa Hahn [9], Mehmet Samur [10,11,12], Nikhil Munshi [12,13], Song Liu[3,15], Philip L. McCarthy[9,15] & Jens Hillengass [1,15✉]

Osteolytic lesions (OL) characterize symptomatic multiple myeloma. The mechanisms of how malignant plasma cells (PC) cause OL in one region while others show no signs of bone destruction despite subtotal infiltration remain unknown. We report on a single-cell RNA sequencing (scRNA-seq) study of PC obtained prospectively from random bone marrow aspirates (BM) and paired imaging-guided biopsies of OL. We analyze 148,630 PC from 24 different locations in 10 patients and observe vast inter- and intra-patient heterogeneity based on scRNA-seq analyses. Beyond the limited evidence for spatial heterogeneity from whole-exome sequencing, we find an additional layer of complexity by integrated analysis of anchored scRNA-seq datasets from the BM and OL. PC from OL are characterized by differentially expressed genes compared to PC from BM, including upregulation of genes associated with myeloma bone disease like *DKK1*, *HGF* and *TIMP-1* as well as recurrent downregulation of *JUN/FOS*, *DUSP1* and *HBB*. Assessment of PC from longitudinally collected samples reveals transcriptional changes after induction therapy. Our study contributes to the understanding of destructive myeloma bone disease.

[1] Department of Medicine, Roswell Park Comprehensive Cancer Center (Roswell Park), Buffalo, NY, USA. [2] Department of Hematology, Cell therapy and Hemostaseology, University Hospital Leipzig, Leipzig, Germany. [3] Department of Biostatistics and Bioinformatics, Roswell Park, Buffalo, USA. [4] Department of Diagnostic Radiology, Roswell Park, Buffalo, USA. [5] Clinical Cytogenetics Laboratory, Department of Pathology and Laboratory Medicine, Roswell Park, Buffalo, USA. [6] Department of Immunology, Roswell Park, Buffalo, USA. [7] Flow and Image Cytometry, Department of Pathology and Laboratory Medicine, Roswell Park, Buffalo, USA. [8] Genomics Shared Resources, Roswell Park, Buffalo, USA. [9] Transplant and Cellular Therapy Program, Department of Medicine, Roswell Park, Buffalo, USA. [10] Department of Data Sciences, Dana Farber Cancer Institute, Boston, MA, USA. [11] Department of Biostatistics, Harvard T.H. Chan School of Public Health, Boston, MA, USA. [12] Department of Medical Oncology, Dana Farber Cancer Institute, Harvard Medical School, Boston, MA, USA. [13] VA Boston Healthcare System, Boston, MA, USA. [14]These authors contributed equally: Maximilian Merz, Almuth Maria Anni Merz, Jie Wang, Lei Wei. [15]These authors jointly supervised: Song Liu, Philip L. McCarthy, Jens Hillengass. ✉email: maximilian.merz@medizin.uni-leipzig.de; jens.hillengass@roswellpark.org

Multiple Myeloma (MM) is a heterogeneous disease with survival ranging from months to decades[1]. Malignant plasma cells (PC) for histopathology and genetic assessment are isolated from iliac crest bone marrow aspirates in routine practice. However, PC are not homogeneously distributed within the bone marrow. Osteolytic lesions (OL) are areas of circumscribed bone loss caused by malignant PC infiltration. While OL can be visualized by positron emission computed tomography (PET/CT) in up to 80% of patients, their underlying biology remains to be clarified. Some patients show subtotal PC infiltration of the bone marrow in the iliac crest without signs of bone destruction while in the same patients, PC cause bone disease in distant locations such as the vertebral bodies. Therefore, OL might represent regions of increased infiltration as well as areas containing biologically different PC.

Efforts have been made to classify MM patients based on copy number changes[2], mutational burden[2] and gene expression profiling (GEP)[3–6]. Recently, a retrospective multi-region whole exome sequencing (WES) study showed for the first time spatial genomic heterogeneity of paired samples from random bone marrow aspirates and distant lesions[7]. Since focal lesions before and after therapy are associated with adverse outcome[8], site-specific high-risk PC populations might be responsible for treatment resistance and relapse. Therefore, sampling PC solely from the iliac crest might not be representative due to intra-patient spatial heterogeneity.

Furthermore, cancers are not composed by an aggregation of genetically identical cells, and bulk tissue sequencing might obscure biologically relevant differences between cells. Among several emerging technologies to interrogate tumors at a single-cell resolution, single-cell RNA sequencing (scRNA-seq) can identify treatment resistant clones and subpopulations responsible for metastatic spread in several human cancers[9–12]. The first scRNA-seq study in MM examined 20,568 PC from bone marrow samples from 29 patients with a variety of plasma cell disorders. This analysis demonstrated PC heterogeneity in MM and identified circulating tumor cells as well as measurable residual disease (MRD) after therapy[10]. More recently, scRNA-seq was used to study asymptomatic, symptomatic and relapsed patients as well as modes of resistance to chimeric antigen receptor T-cell therapy[13–16].

In this work, we conduct a scRNA-seq analysis of 148,630 freshly purified PC obtained prospectively from random bone marrow aspirates and paired imaging-guided biopsies of OL in 10 patients with symptomatic MM. We demonstrate that scRNA-seq of PC from OL is feasible in a prospective clinical trial. Based on single cell transcriptomics, we observe inter- as well as intra-patient heterogeneity. While we show limited spatial heterogeneity based on WES, scRNA-seq identifies significant differences between both locations. Assessment of PC sampled after induction therapy shows transcriptional changes compared to baseline findings. Our study adds an additional layer of complexity to spatial heterogeneity in MM and contributes to the understanding of myeloma bone disease.

## Results

### Single cell RNA sequencing of plasma cells from guided biopsies of osteolytic lesions and corresponding bone marrow identifies inter-patient heterogeneity.

We implemented a translational workflow (Fig. 1) to obtain and purify viable PC from OL and corresponding BM samples from patients with newly diagnosed or relapsed MM. The bone marrow biopsies from the iliac crest (bone marrow sample, BM) were obtained and processed at the same time as OL biopsies since differences in sample processing times might cause changes in MM gene expression[17]. We performed scRNA-seq on paired samples from 10 patients (7 with newly diagnosed and 3 relapsed/refractory MM) with one patient having 2 OL biopsied. In three patients with NDMM, we obtained subsequent samples after induction therapy. Patient characteristics are summarized in Table 1, treatment is summarized in supplemental Table 1. With the exception of patient RRMM01 with para-medullary spread from an OL of the right clavicle, all samples were acquired from intra-medullary lesions. There were no significant differences between both locations regarding the purity of isolated and sequenced PC, underlining the feasibility of our protocol and comparability of paired samples. In total, 94.8% of cells from OL ($n = 70,036$) and 95.7% of cells from BM ($n = 71,580$) were PC (Fig. 1).

Clustering of 148,746 single cells from BM and OL (median 7712 cells/sample) created a map of distinct populations based on transcriptomes from individual patients and locations (Fig. 2A, B). Cells from individual patients clustered together in both, BM and OL (Fig. 2C). This was also observed when merging cells from both locations (Fig. 2D) which demonstrated that inter-patient heterogeneity outweighed spatial heterogeneity and determined clustering patterns of malignant PC. Clusters with overlapping cells from different patients were later identified as few contaminating, non-PC (Fig. 2E).

To investigate inter-patient heterogeneity, we identified marker genes for malignant PC clusters from each individual patient (Fig. 3A, B). As expected, the genes for cyclin D1-3 (CCND1-3) were preferentially expressed in patients with IgH translocations (NDMM02, NDMM04 and NDMM06) as detected by FISH (Table 1). Furthermore, genes associated with myeloma bone disease (DKK1 and FRZB), cytokine signaling (IFI27 and IL6R) and EDNRB were identified as marker genes for PC clusters from individual patients. All of the aforementioned genes were previously described as characteristic genes elevated in the molecular subtypes of MM that were derived from bulk GEP[18,19]. Besides these known marker genes, we identified STMN1 to be preferentially expressed in small subsets of malignant PC (Fig. 3B). No significant differences for the identified marker genes were found between BM and OL.

Since we observed vast inter-patient heterogeneity, we performed a gene set enrichment analysis (GSEA) using the curated MM subtype gene sets from the Molecular Signature Database (MSigDB) to assess whether our findings are consistent with the molecular classification of MM that was established with GEP. In agreement with the single gene analysis, gene-set analysis showed that patients could be grouped according to the molecular classification of MM. The respective classification differentiates 7 MM subtypes influenced by the presence of genetic lesions such as translocations and a hyperdiploid karyotype as well as low-incidence of bone disease or increased expression of proliferation-associated genes. Beyond the genetic and phenotypic differences between the groups, the molecular classification is also of prognostic significance with adverse outcome for patients in the proliferation (PR) group. We demonstrate that scRNA-seq reveals inter-patient heterogeneity and identifies MM subtypes that are consistent with the well-established molecular classification of MM. For example, patient RRMM02 in the low bone disease (LB) group showed higher expression of IL6R and EDNRB and NDMM02 in the MAF group (MF) showed higher levels of CCND2 and lower levels of DKK1 (Fig. 3C).

### Intra-patient heterogeneity based on scRNA-seq.

After investigating inter-patient heterogeneity using scRNA-seq data, we aimed at deciphering intra-patient heterogeneity on the transcriptional level. We performed clustering (Fig. 4A) and differential expression analysis (Fig. 4B), GSEA (Fig. 4C), and inferred

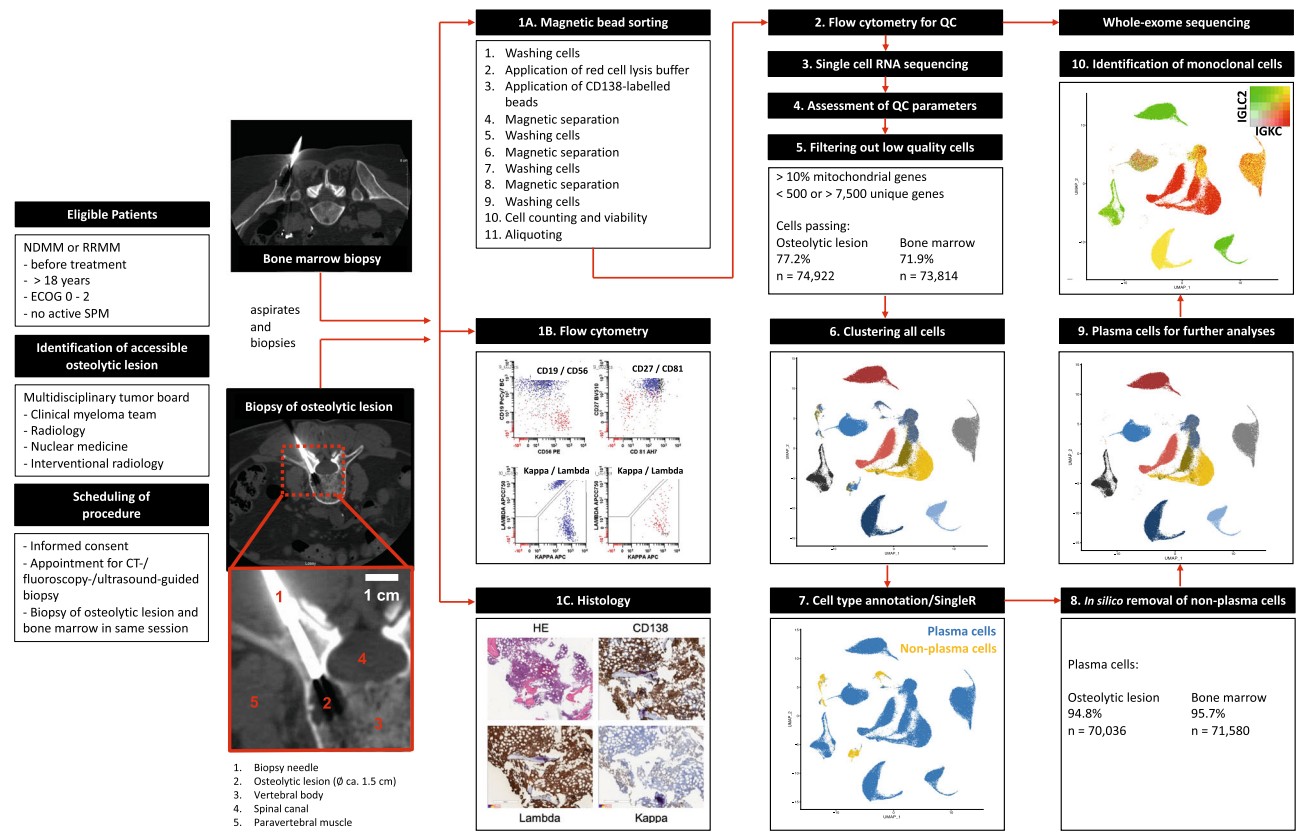

**Fig. 1 Acquisition of viable plasma cells from imaging-guided biopsies of osteolytic lesions and bone marrow.** Eligible patients with newly diagnosed (NDMM) or relapsed/refractory multiple myeloma (RRMM) were discussed in a multidisciplinary tumor board before the initiation of therapy. Patients consented to an imaging-guided biopsy of an osteolytic lesion (OL) in addition to the routine diagnostic bone marrow aspirate from the iliac crest (BM). To minimize changes in gene expression due to delayed sample processing, aspirates of the BM and OL were performed and analyzed on the same day as follows: (1A) Plasma cells (PC) were isolated using CD138 positive selection; Clinical samples were investigated by flow cytometry (1B) and immunohistochemistry (1C) to confirm the diagnosis[2] Fluorescence-activated cell sorting (FACS) was performed for quality control (QC)[3]; cells were subjected to single cell RNA sequencing (scRNA-seq). The rest of the PC were frozen at −80 °C and later analyzed by whole exome sequencing (WES). After assessment of QC parameters[4], low quality cells were filtered out[5]. Standard Seurat workflow was performed to cluster all cells[6]. After identification of contaminating non-malignant PC (yellow, PC in blue)[7] the respective cells were removed in silico[8] for further analyses[9]. Malignant PC were identified by restricted heavy and light chain expression[10]. QC parameters were comparable between OL and BM and ca. 95% of captured and sequenced cells from BM and OL were PC underlining feasibility of our workflow.

CNVs (Fig. 4D) from PC for each individual patient. Specifically, we looked for genes that were recurrently differentially expressed in PC clusters and that have been associated to play a role in the pathogenesis and prognostication of MM. A summary of the differential expression analysis for each individual patient is available at Supplemental Data 1.

We identified clusters of malignant PC characterized by the overexpression of genes encoding the microtubule-associated proteins *STMN1* and *TUBA1B* (Fig. 4A, B). GSEA showed that gene sets associated with proliferation, oxidative phosphorylation and MYC targets were significantly enriched in these clusters (Fig. 4C). However, no significant differences in CNVs were identified in STMN1-positive clusters compared to the remaining malignant PC (Fig. 4D).

The respective clusters were found in all patients and locations (Fig. 4E, F), which was also confirmed by non-negative matrix factorization (Supplemental Fig. 1). No significant differences in the number of cells expressing *STMN1* were found between OL and BM, except for patient RRMM01 with para-medullary disease (Fig. 4G). Since GSEA demonstrated an association with pathways connected to proliferation, we assigned cell cycle scores to PC (Fig. 4H) and found that cells mapped to clusters residing predominantly in synthesis (S)-Phase compared to the remaining

malignant PC in G1-Phase (Fig. 4H). To ensure that differences in cell cycle stages did not introduce bias into further analyses on spatial heterogeneity, we compared both locations and did not find significant differences between OL and BM (Fig. 4I).

The number of PC in S-Phase and the proliferation index are well-established factors for adverse outcome in MM[18]. *STMN1* is among the 15 genes associated with high-risk disease in the GEP score identified by the Integroupe Francophone du Myelome[4]. Higher expression levels of *STMN1* and *TUBA1B* are associated with shorter progression-free and overall survival in the CoMMpass dataset of the Multiple Myeloma Research Foundation (https://research.themmrf.org/, Supplemental Fig. 2). Furthermore, *STMN1* was associated with refractory and resistant disease in a recent scRNA-seq study[20]. To further objectify the potential prognostic significance of the respective clusters, we mapped expression of genes in the University of Arkansas for Medical Science 17 high-risk gene score (UAMS17) onto single PC and found an overlap between *STMN1* expression and higher UAMS17 gene expression levels (Fig. 4J).

This underlines that scRNA-seq characterized intra-patient heterogeneity and recurrent subclusters of transcriptionally different PC among different patients and locations. We provide a link between well-established factors for adverse outcome. Since

**Table 1 Baseline characteristics and quality measures of single cell RNA sequencing.**

| Patient | Sex | Disease status | Subtype | Location | PMD | Histology % plasma cells | FACS % plasma cells after sorting | viability | scRNA-seq cells sequenced | cells passed QC | IgG 700–1600 mg/dl | IgA 70–390 mg/dl | IgM 50–230 mg/dl | kappa 3–19 mg/l | lambda 6–26 mg/l | FLC | Hb 12.5–15.5 g/dl | Crea 0.52–1.04 mg/dl | Calcium 8.4–10.2 mg/dl | Albumin 3.5–5.0 g/dl | beta2 0.8–2.34 mg/l | LDH 313–618 IU/l | FISH |
|---|---|---|---|---|---|---|---|---|---|---|---|---|---|---|---|---|---|---|---|---|---|---|---|
| NDMM01 | male | newly diagnosed | IgG kappa | bone marrow L2 | n | 10 | 96 | 92 | 4365 | 2227 | 1006 | 180 | 62 | 45 | 13 | 3 | 15.3 | 1.09 | 9.9 | 4.6 | 2.29 | 361 | normal |
| NDMM02 | male | newly diagnosed | IgG lambda | bone marrow iliac crest (L) | n | 20<br>70 | 96<br>97 | 90<br>98 | 8086<br>9709 | 7788<br>8753 | 4053 | 0 | 0 | 11 | 2482 | 226 | 11.9 | 1.44 | 8.7 | 3.9 | | | HD, del3, del5, IgH unknown |
| NDMM03 | female | newly diagnosed | IgG kappa | bone marrow L3 | n | 60 | 99 | 94 | 11578 | 10303 | 1407 | 218 | 89 | 536 | 25 | 21 | 14.3 | 0.78 | 9.6 | 4.8 | 3.54 | 427 | normal |
| NDMM04 | female | newly diagnosed | lambda | bone marrow sacrum | n | 10<br>10 | 79<br>91 | 89<br>97 | 10029<br>10266 | 9184<br>8810 | 478 | 0 | 0 | 11 | 1293 | 118 | 12.9 | 0.85 | 9.7 | 4.1 | 4.25 | 647 | IgH unknown |
| NDMM05 | female | newly diagnosed | IgG kappa | bone marrow iliac crest (R) | n | 50<br>25 | 78<br>95 | 96<br>89 | 10012<br>13367 | 8563<br>12825 | 3825 | 70 | 59 | 559 | 7 | 80 | 10.4 | 0.70 | 9.7 | 4.4 | 2.27 | 472 | HD (both) |
| NDMM06 | female | newly diagnosed | IgA kappa | bone marrow T11<br>bone marrow L4 | n | 60<br>80<br>80<br>95 | 100<br>100<br>100<br>100 | 100<br>97<br>97<br>99 | 9853<br>9988<br>9268<br>14135 | 6700<br>8207<br>7549<br>6849 | 0 | 3171 | 25 | 59 | 1 | 59 | 9.8 | 0.60 | 9.9 | 4.3 | 4.09 | 353 | del3, del16, del7, gain15, t(11;14) |
| NDMM07 | male | newly diagnosed | IgG kappa | bone marrow iliac crest (L) | n | 95<br>95 | 91<br>na | 99<br>81 | 9921<br>8297 | 6092<br>7865 | 9338 | 52 | 0 | 3343 | 6 | 557 | 7.2 | 2.67 | 10.7 | 2.8 | 18.95 | 714 | del1p, HD |
| RRMM01 | male | relapsed | IgA kappa/lambda | bone marrow clavicle (R) | y | 5 | 91 | 81 | 8297 | 7865 | 702 | 222 | 36 | 15 | 64 | biclonal | 11.0 | 0.84 | 8.6 | 3.7 | | | gain1, t(MYC) |
| RRMM02 | male | relapsed | IgG lambda | bone marrow iliac crest (R) | n | 100<br>60 | na<br>87 | 92<br>90 | 1742<br>11870 | 1383<br>6925 | 297 | 0 | 0 | 2 | 1 | 1 | 12.5 | 0.94 | 9.5 | 4.3 | | | HD |
| RRMM03 | male | relapsed | IgG kappa | bone marrow iliac crest (R) | n | 60<br>15<br>15 | 94<br>97<br>99 | 85<br>87<br>90 | 3329<br>14746<br>18401 | 2110<br>7712<br>9594 | 730 | 0 | 0 | 72 | 6 | 12 | 11.5 | 0.76 | 8.6 | 3.9 | 2.18 | 508 | del6, del7 |

*PMD paramedullary disease (n = no, y = yes), FACS fluorescence-activated cell sorting, scRNA-seq single cell RNA sequencing, QC quality control, Hb hemoglobin, Crea Creatinine, beta2 beta-2 microglobulin, LDH lactate dehydrogenase, FISH Flourescence in situ hybridization, HD hyperdiploid karyotype.*

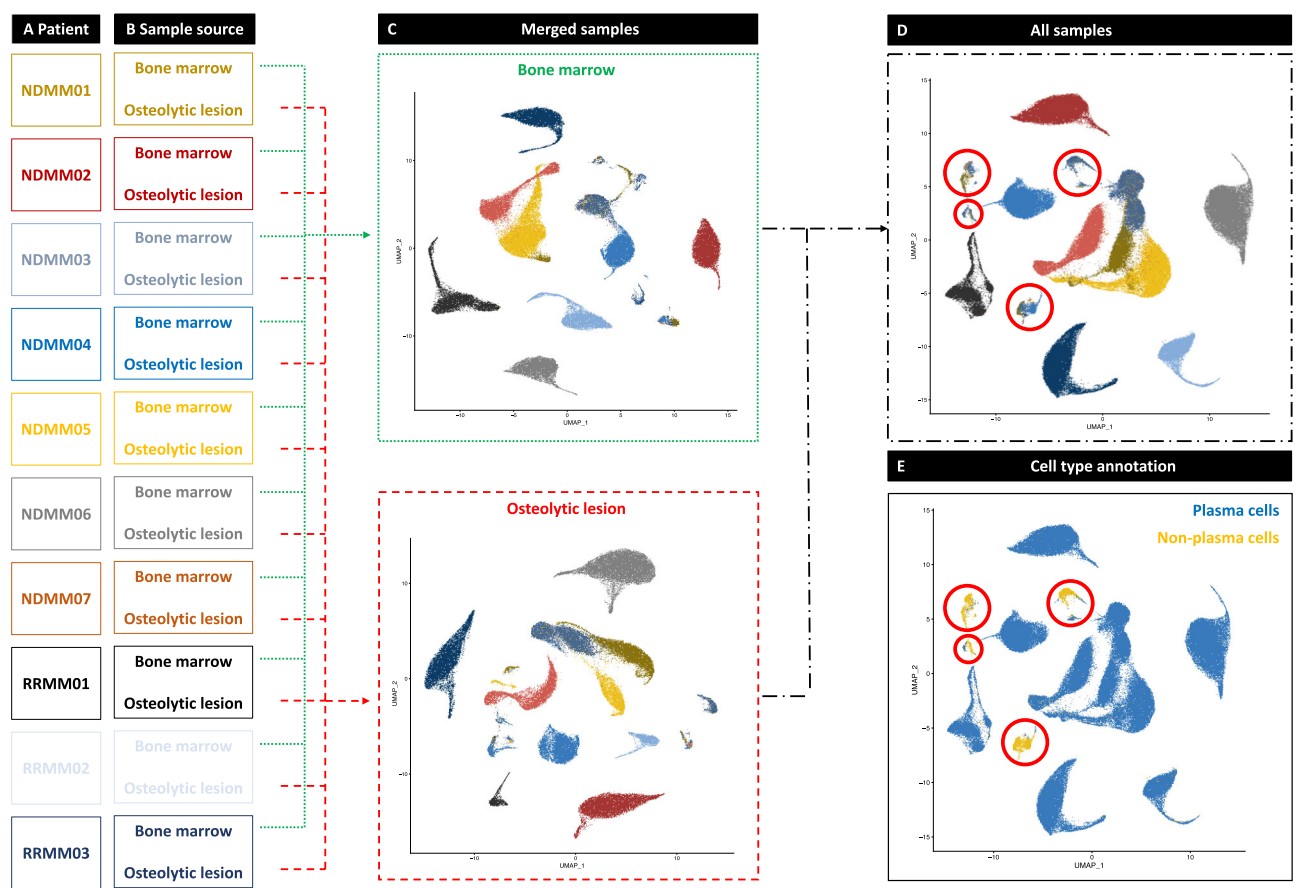

**Fig. 2 Clustering cells from different locations and patients.** To investigate whether malignant plasma cells (PC) from different patients **A** or locations **B** clustered jointly on Uniform Manifold Approximation and Projection (UMAP) plots, we first merged samples separately from bone marrow and osteolytic lesion **C**. PC from individual patients are depicted by specific colors. Merging the entire dataset revealed that cells from individual patients clustered together **D**. The only regions with overlapping cells from different patients were identified as few contaminating, non-PC **E**. This provided evidence that inter-patient heterogeneity was more significant compared to spatial heterogeneity.

the respective clusters accounted only for a small number of malignant PC in each individual patient they would have been missed by bulk sequencing (Supplemental Fig. 2).

**Limited evidence for spatial heterogeneity in patients with intra-medullary lesions from whole-exome sequencing.** Next, we aimed at characterizing spatial heterogeneity. Whole-exome sequencing (WES) has been utilized to investigate spatial genomic heterogeneity in MM based on banked, frozen PC in a retrospective study[7]. To explore whether scRNA-seq on freshly isolated PC could reveal another layer of complexity of spatial heterogeneity, we performed WES on all paired samples. The comparison of PC with matched normal germline cells identified a total of 1103 somatic mutations, including 1063 SNVs and 40 Indels (Fig. 5A). Among these somatic mutations, 665 were predicted to cause an amino acid alteration, 72 were truncating and 366 were silent mutations. Supplemental Data 2 gives an overview of the individual variant calls from the WES analysis. For each patient, we quantified the similarity between the BM and OL by calculating the Jaccard score, defined as the ratio between shared mutations and all mutations. Patient NDMM03 was excluded from calculations since only limited numbers of malignant PCs were captured from the BM.

In 8 out of 10 patients (NDMM02 to NDMM07 and RRMM02 and RRMM03), the percentage of shared mutations between the bone marrow and OL were ~80% or higher, suggesting the BM and OL were highly consistent (Fig. 5A, B). In the patient with 2

OL (NDMM06), the Jaccard scores among all 3 locations (BM and two OL) were above 97% with a clonal *TP53* mutation present in all three locations.

WES revealed relevant spatial heterogeneity in 2 out of 10 patients: For patient NDMM01, 75% of all mutations were shared between the bone marrow and OL, and 25% of mutations were only present in the OL including a *BRAF V600E* mutation. For patient RRMM01 with an OL of the right clavicle with paramedullary spread, only 20% of all mutations were shared, with 24% of the mutations found only in the bone marrow, and 56% of the mutations unique to the OL. Two distinct *BRAF* mutations were detected: *V600E* was found in BM, and a different activating Class 2 *BRAF* mutation (*G469R*) in the OL. The latter mutation causes resistance to the BRAF inhibitor vemurafenib[21]. Furthermore, we detected an additional *NRAS* mutation (*G12D*) in the OL. *NRAS* mutations have been associated to drive spatially divergent resistance to vemurafenib in BRAFmut MM[22]. These examples occurred in the absence of exposure to vemurafenib and demonstrate that treatment with a BRAF inhibitor would most likely be ineffective against PC from different locations these patients.

**Single cell RNA sequencing revealed another layer of complexity and links site-specific gene expression to the development of osteolytic lesions.** After analyzing WES from malignant PC from the OL and BM, we investigated whether the observed similarities between both locations are also reflected by scRNA-seq

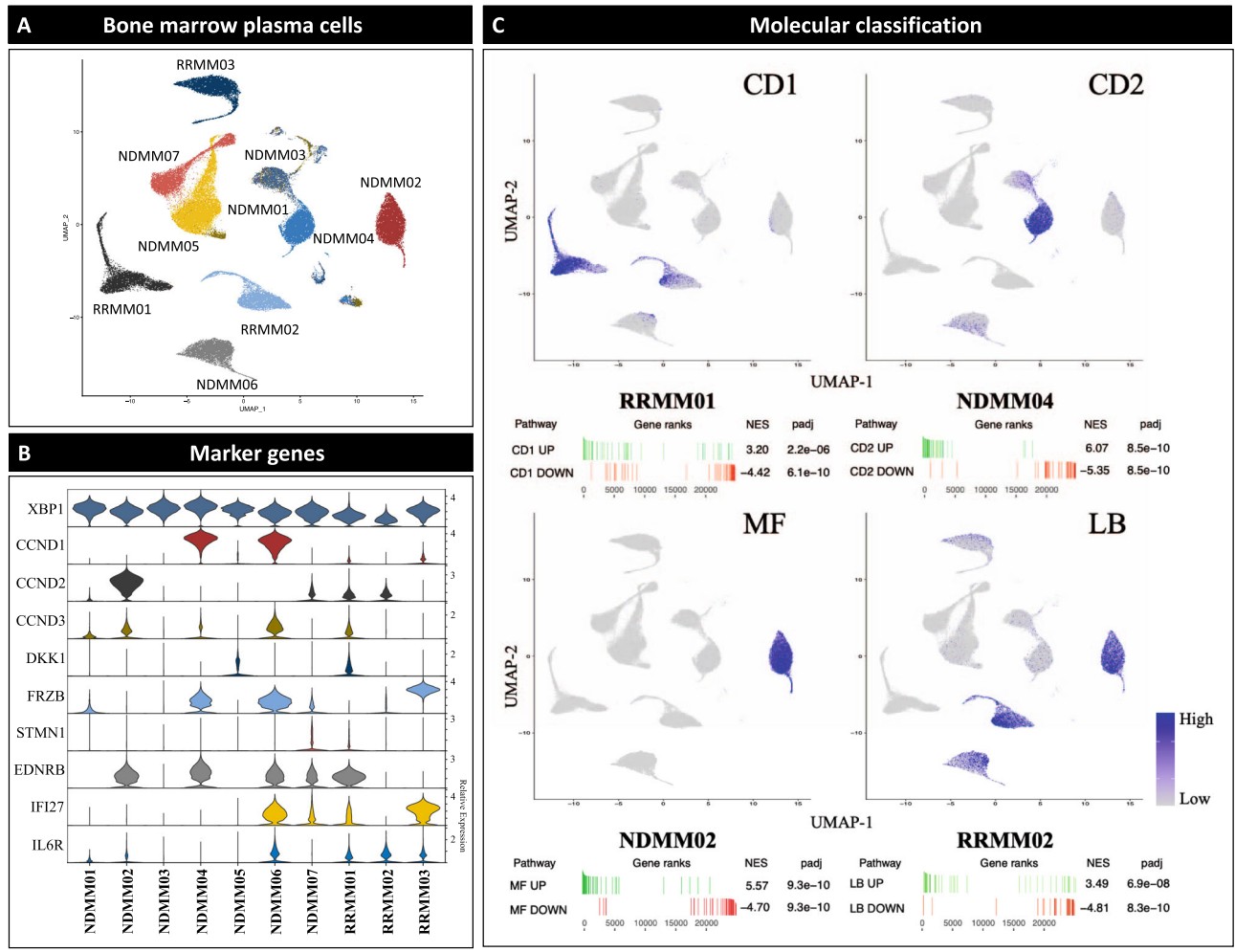

**Fig. 3 Single cell RNA sequencing demonstrates inter-patient heterogeneity and identifies molecular subgroups of multiple myeloma. A** To further characterize inter-patient heterogeneity based on scRNA-seq of bone marrow plasma cells (PC) we identified marker genes for the patient-specific single PC clusters. **B** Violin plots showing the identified marker genes for each individual patient. The width of the violin representing the number of cells expressing the respective gene with the relative expression being plotted on the y-axis. *XBP1* was included in the top row as positive control to show the extend of the entire PC population. *CCND1-3* were marker genes in patients with IgH-translocations (NDMM02/04/06) and homogeneously expressed in all PC. Other marker genes associated with bone disease (*DKK1*, *FRZB*), cytokine signaling (*IFI27*, *IL6R*) and *EDNRB* have been described in bulk gene expression profiling (GEP) studies. **C** Gene set enrichment analysis (GSEA) to classify patients according to the molecular subtypes published by the Arkansas group[18]. Gene sets are available in the Molecular Signature Database (MSigDB). Examples are shown for the subgroups CD1 (RRMM01), CD2 (NDMM04), MF (NDMM02) and LB (RRMM02). GSEA for up- (green lines) and downregulated genes (red lines) were performed. The AddModuleScore() command from Seurat was used to calculate the average expression for the top 50 upregulated genes in the molecular subcategory and visualize findings in a FeaturePlot. Relative expression levels ranged from low (in gray) to high (in purple). ScRNA-seq characterized inter-patient heterogeneity and classified individual patients based on their single PC transcriptomes into molecular subgroups of multiple myeloma. NES = normalized enrichment score. Padj = adjusted *p*-value estimation based on an adaptive multi-level split Monte-Carlo scheme.

data. Therefore, average gene expression of PC from the OL and BM were correlated to each other (Fig. 5B). In agreement with WES findings, we found a high concordance of average gene expression between both locations. However, in every patient, outliers in both directions were observed (Fig. 5B).

To identify genes that are differentially expressed in PC from both conditions, we performed an integrated analysis after anchoring datasets from OL and BM for each individual patient as described before[23]. This process identifies pairwise correspondences between PC from different origins—called *anchors*—to transform datasets into a shared space. By aligning PC from BM and OL we were able to directly compare single PC gene expression from both locations. After applying the integration procedure, malignant PC were robustly detected in all datasets and the same PC clusters were identified in BM and OL. In all

patients, we were able to find marker genes that were differentially expressed in malignant PC from OL when compared to malignant PC from BM. Overall, 1140 genes were identified that were differentially expressed between OL and BM (Fig. 5C). Genes that have been associated with the development of myeloma bone disease such as *DKK1*, *HGF* (Fig. 5C) and *TIMP-1* (Fig. 5C)[24,25] were among the markers with higher expression levels in OL (Fig. 5C, D). Furthermore, in agreement with the first scRNA-seq analysis in MM[10], we found *LAMP5* to be upregulated in PC from the OL (Fig. 5C, D).

Genes that were recurrently downregulated in PC from OL were *JUN* (Fig. 5C) and *FOS* (Fig. 5C, D) (6 of 10 patients), dual specificity phosphatase 1 (*DUSP1*, Fig. 5C, 5 of 10 patients) and hemoglobin beta chain (*HBB*, 3 of 10 patients) (Fig. 5C, D). Importantly, no somatic mutations were detected in the

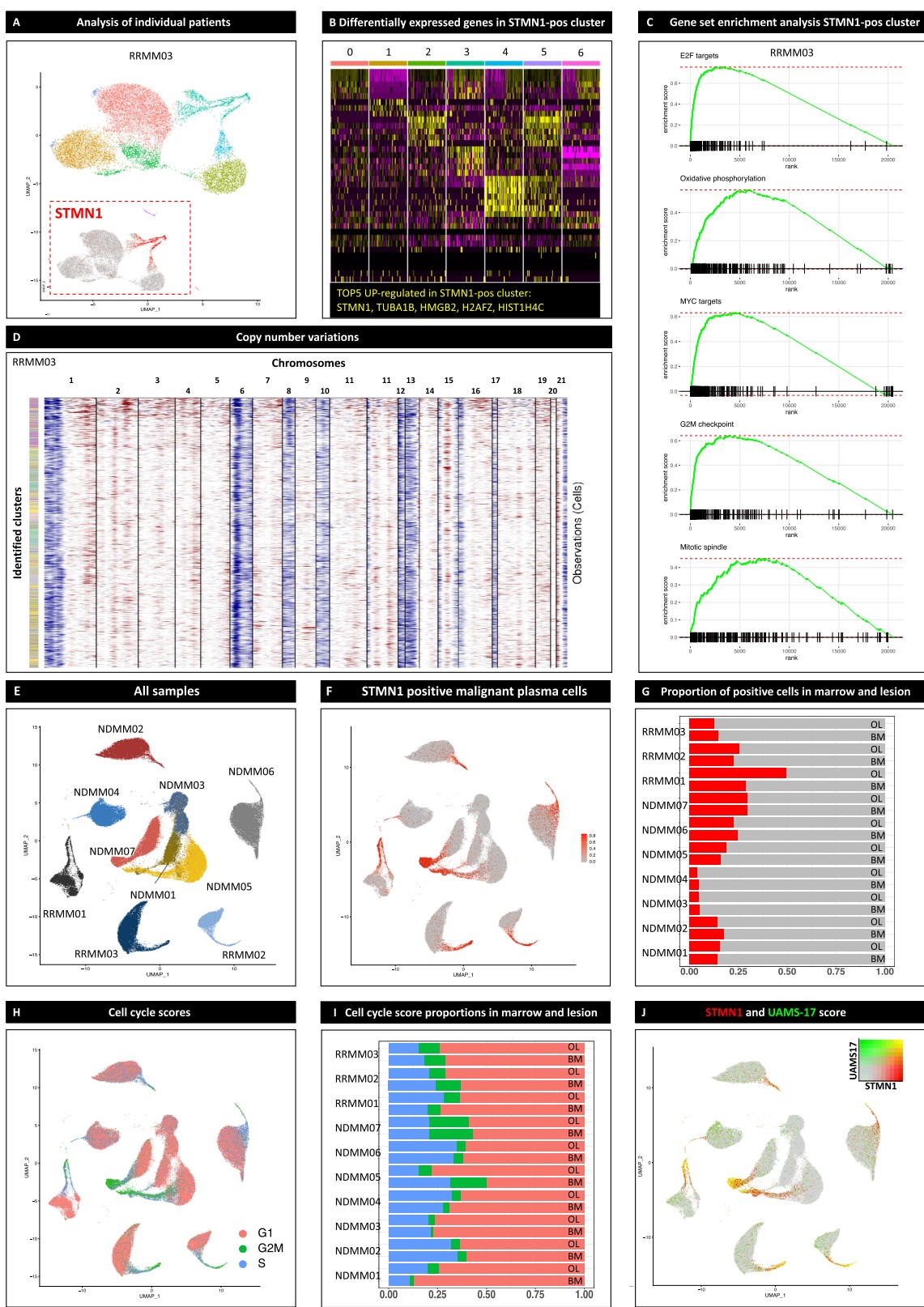

respective genes in PC from the BM and OL. Downregulation of *JUN/FOS*, *DUSP1* and *HBB* has been connected to extramedullary spread of MM in the past[26]. Furthermore, *JUN/FOS* are linked to the malignant transformation of B-cells[27] and dissemination of clonal PC in a preclinical model[28]. GSEA confirmed that downregulation of genes in pathways associated with normal B-cell was common in PC from the OL (Supplemental Fig. 3).

Additionally, lower expression levels of genes encoding for the non-restricted light (e.g., *IGLC2/3* in NDMM01, Fig. 5C) and heavy chain (e.g., *IGHM* in NDMM03, Fig. 5C) were observed in PC from OL. Downregulation of the affected heavy chain was observed in PC from the OL in RRMM01 and RRMM03 providing a link between evolving disease and the rare phenomenon of light chain escape that can be observed in

**Fig. 4 Intra-patient heterogeneity based on single plasma cell transcriptomes and inferred copy number variations.** After the identification of inter-patient heterogeneity, we performed clustering **A**, differential expression **B** as well as gene set enrichment analysis **C** and inferred copy number variations (CNV, **D**) for every individual patient to characterize intra-patient heterogeneity. **A–D** provides an example of this process for patient RRMM03 with relapsed disease. We looked specifically for recurrent marker genes that characterized distinct subsets of malignant plasma cells (PC) across multiple patients and locations. We detected subclusters characterized by the overexpression of genes encoding the microtubule-associated proteins *STMN1* and *TUBA1B* **A**, **B**. Gene set enrichment analysis (GSEA) revealed an upregulation of pathways associated with proliferation and oxidative phosphorylation in the respective clusters compared to the remaining malignant PC **C**. No significant differences in CNVs (gains in red, losses in blue) were observed in the *STMN1*-positive clusters **D**. Next, we investigated whether the respective *STMN1*-positive clusters can be detected in all patients and locations **E**. *STMN1*-positive cells were found in every individual patient **F** with no significant differences between osteolytic lesion (OL) and bone marrow (BM) in the number of cells expressing *STMN1* **G**. Only patient RRMM01 with para-medullary spread of the disease harbored more *STMN1*-pos cells in OL. Since GSEA provided evidence that the respective cells were proliferatively active, we assigned cell cycle scores using the markers preloaded in Seurat. G1 phase in red, G2M phase in green, S phase in blue **H**. It was confirmed that the majority of the cells in STMN-1 positive clusters were in synthesis (S-)phase **I** and no differences in cell cycle scores were found between OL and BM **J**. Proliferation and the number of S-Phase PC are well-established risk factors for adverse outcome and higher expression of *STMN1* and *TUBA1B* were associated with adverse outcome in the CoMMpass dataset (Supplemental Fig. 2). The potential prognostic implications were supported by higher expression of genes in the UAMS-17 high-risk score. Relative expression of UAMS-17 genes ranging from low in gray to high in green and *STMN1* from low in gray to high in red. High expression of both UAMS-17 score and *STMN1* in the respective cells depicted by yellow color **J**. These results demonstrate that scRNA-seq identifies intra-patient heterogeneity and characterizes single PC with different risk profiles.

heavily pretreated RRMM and PMD. In agreement with this finding, both patients showed low secretory activity in serum (Table 1).

In patient RRMM01 with a PC tumor with para-medullary spread, we found the most significant differences between PC from BM and OL (Fig. 6). While same PC clusters were identified in both locations (Fig. 6A) significant differences in gene expression (Fig. 6B, C) and inferred CNVs (Fig. 6D) were detected among different PC clusters. Remarkably, trajectory inference revealed that the cluster showing over-expression of OL-associated genes *LAMP5* and *HGF* underwent the largest transcriptional change compared to the remaining PC. Besides the respective genes that were also up-regulated in PC from OL in other patients, we identified the gene for zinc-alpha2-glycoprotein (*AZGP1*) to be downregulated in the OL (Fig. 6B). *AZGP1* is a known tumor suppressor gene and its loss causes epithelial-to-mesenchymal transition (EMT)[29]. While *AZGP1* expression was virtually absent in PC from the OL, it was homogeneously expressed in the majority of PC from the BM. However, a small cluster of PC with lower *AZGP1* expression levels was identified in the BM (Fig. 6C). GSEA analysis confirmed that genes associated with EMT were enriched in the respective cluster compared to the rest of malignant PC (Fig. 6C). Furthermore, gene sets connected to proliferation and oxidative phosphorylation were downregulated in these PC. This confirms the hypothesis from pre-clinical animal models that hypoxia-driven, EMT-like processes connected to chemoresistance and decreased proliferation drive extramedullary spread of MM cells[30]. It can be hypothesized that the small PC cluster might have occurred latest in the developmental process and could have given rise to the OL in the right clavicle. These findings need to be interpreted with caution since we were only able to study a single patient with para-medullary disease, while other patients had strictly intra-medullary disease.

Our results show that scRNA-seq adds another layer of complexity compared to WES to spatial heterogeneity in MM. The scRNA-seq data allow us to link site-specific gene expression to the development of myeloma bone disease and identify subclusters that might be the origin for OL.

**Single cell RNA sequencing characterizes measurable residual disease and transcriptional changes after therapy.** In three patients (NDMM01, NDMM03, and NDMM06) we collected samples after 4 cycles of induction therapy. While in NDMM01 (Fig. 7A) and NDMM06 (Fig. 7C) we performed a regular bone

marrow biopsy to assess residual disease, an imaging-guided biopsy of a residual focal MRI lesion in T8 was biopsied in NDMM03 (Fig. 7B) after treatment. NDMM01 and NDMM03 were in MRD-positive complete remission (CR) after 4 cycles lenalidomide, bortezomib and dexamethasone (RVD). NDMM06 had achieved a partial response (PR) after 4 cycles daratumumab-RVD. Treatment regimen are summarized in supplemental Table 1. Correspondingly, we captured less malignant PC in NDMM01 (Fig. 7A) and NDMM03 (Fig. 7B), while almost the same amount of malignant PC after therapy were isolated and sequenced in NDMM06 compared to the BM and the two OL at primary diagnosis (Fig. 7C).

To identify transcriptional programs that are shared and unshared between samples from all three patients before and after therapy, we used linked interference of genomic experimental relationships (LIGER)[31]. LIGER allows for joint identification of cell types and shared as well as dataset-specific factors between multiple datasets through integrative non-negative matrix factorization.

After the integration process, cells were plotted by treatment status (before/after treatement, Fig. 7D) and identified clusters (Fig. 7E). To confirm that the majority of captured cells represented PC, expression levels of known PC markers were analyzed and no significant differences were found before and after therapy for the respective genes (Fig. 7F). As expected, less malignant PC were captured after treatment (Fig. 7G) with the exception of cluster 0 that represented predominantly PC from patient NDMM06 with a PR (Fig. 7G).

LIGER identified shared and unshared genes between patients before and after therapy (Fig. 7H, I). *DDIT4* (Fig. 7J) and *TXNIP* (Fig. 7K) were downregulated after therapy. Both genes are induced by dexamethasone and involved in glucocorticoid-induced apoptosis providing a link to steroid-refractoriness in the detected residual disease[32,33]. Higher expression levels of *HLA-DRA* and *HLA-DPA1* were observed in malignant PC (Fig. 7I).

*ISG15* was also upregulated in residual PC (Fig. 7L). *ISG15* encodes an ubiquitin-like protein and its expression has been linked to Carfilzomib-resistance in a recent preclinical study[34].

This demonstrates that scRNA-seq characterizes MRD in patients in remission and reveals transcriptional changes consistent with immunomodulatory effects of lenalidomide and drug resistance to proteasome inhibitors. However, our observations were based on only three patients showing feasibility of this methodology to characterize residual cells. Future longitudinal

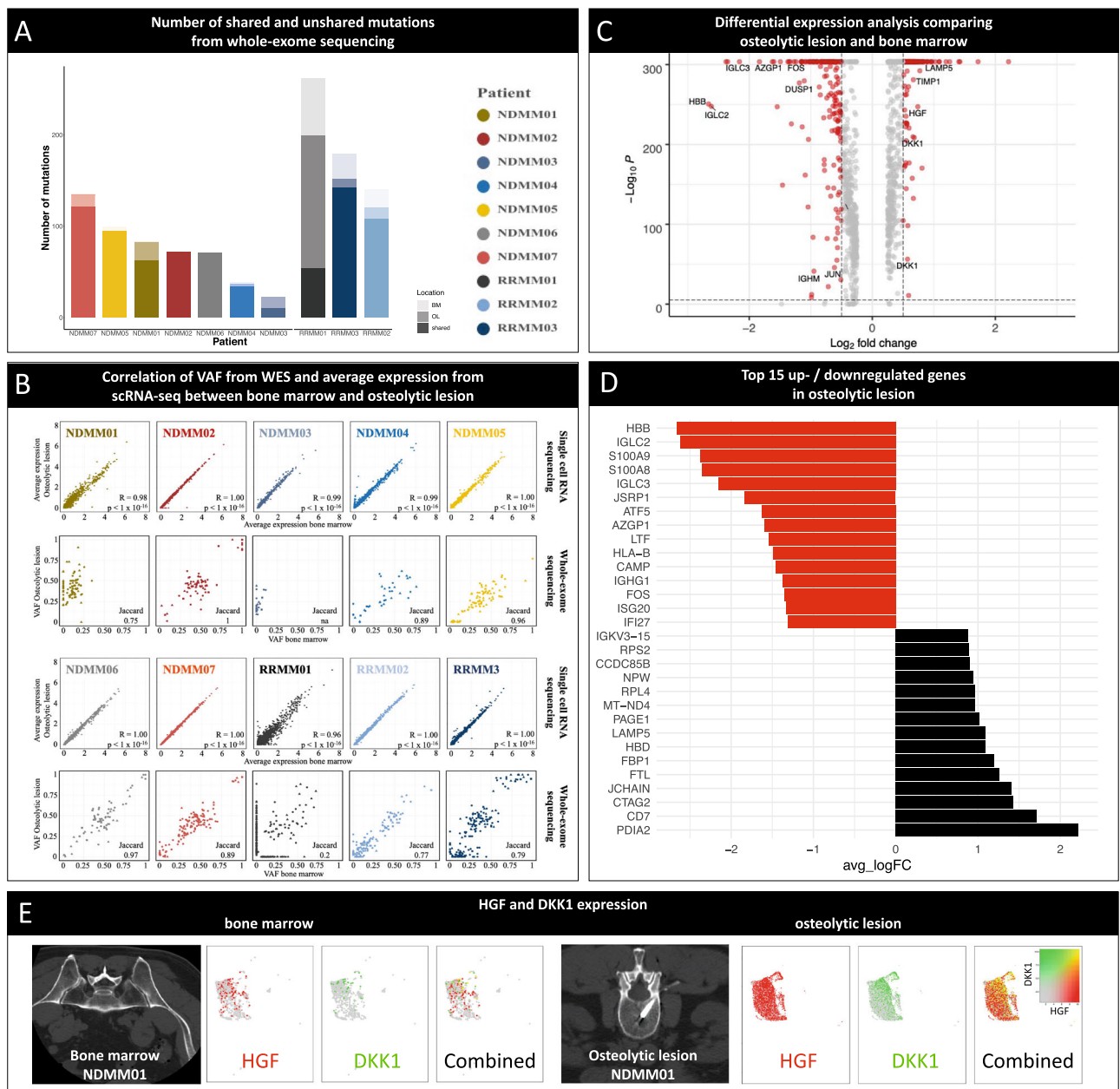

**Fig. 5 Limited evidence for spatial heterogeneity from whole-exome sequencing and significant correlation of gene expression from plasma cells from bone marrow and osteolytic lesion. A** Bar plots showing the number of shared and unshared mutations from whole-exome sequencing (WES) for each individual patient (bone marrow/BM = light colors, osteolytic lesion/OL = medium colors, shared = dark colors). Patients with relapsed/refractory multiple myeloma (RRMM) harbored more mutations compared to newly diagnosed disease (NDMM). **B** First and third rows: Scatter plots for average gene expression from single cell RNA sequencing. Second and fourth rows: Variant allele frequency (VAF) from WES. Results from plasma cells (PC) from the bone marrow (BM) are plotted to x-axis and from the osteolytic lesion (OL) to the y-axis for each individual patient. Jaccard indices were calculated to quantify the overlap between both samples based on WES. In WES scatter plots, effects of the mutations are delineated by different symbols (altering = circle, non-amino acid change = triangle, truncating = square). Only in patient RRMM01 with para-medullary disease (PMD), we found substantial differences as indicated by a Jaccard score of 0.2. Also average gene expression of PC from both location was highly correlated to each other as indicated by two-sided Spearman's correlation coefficients. Nevertheless, outliers in both directions could be detected with scRNA-seq. **C** Differential expression analysis comparing OL and BM identified 1140 significantly up- and downregulated genes in PC from OL. While genes that have been associated with myeloma bone disease were upregulated (e.g., *DKK1*, *HGF*, and *TIMP1*) in OL, *JUN/FOS*, *DUSP1* and *HBB* were recurrently downregulated. Log2 fold change between OL and BM plotted on x-axis, p-values (Bonferroni corrected) derived from two-sided Wilcoxon Rank Sum test on y-axis **D** Top 15 up- and downregulated genes in PC from OL with regards to average log-fold change (avg_logFC). Gene set enrichment analysis demonstrated that genes connected to regular B-cell function were significantly downregulated in OL (Supplemental Fig. 3). **E** Comparison of *HGF* (red) and *DKK1* (green) expression in malignant PC from BM (left side of the panel) and OL (right side of the panel) in patient NDMM01. The number of cells expressing both genes (yellow) as well as expression levels were lower in BM compared to OL.

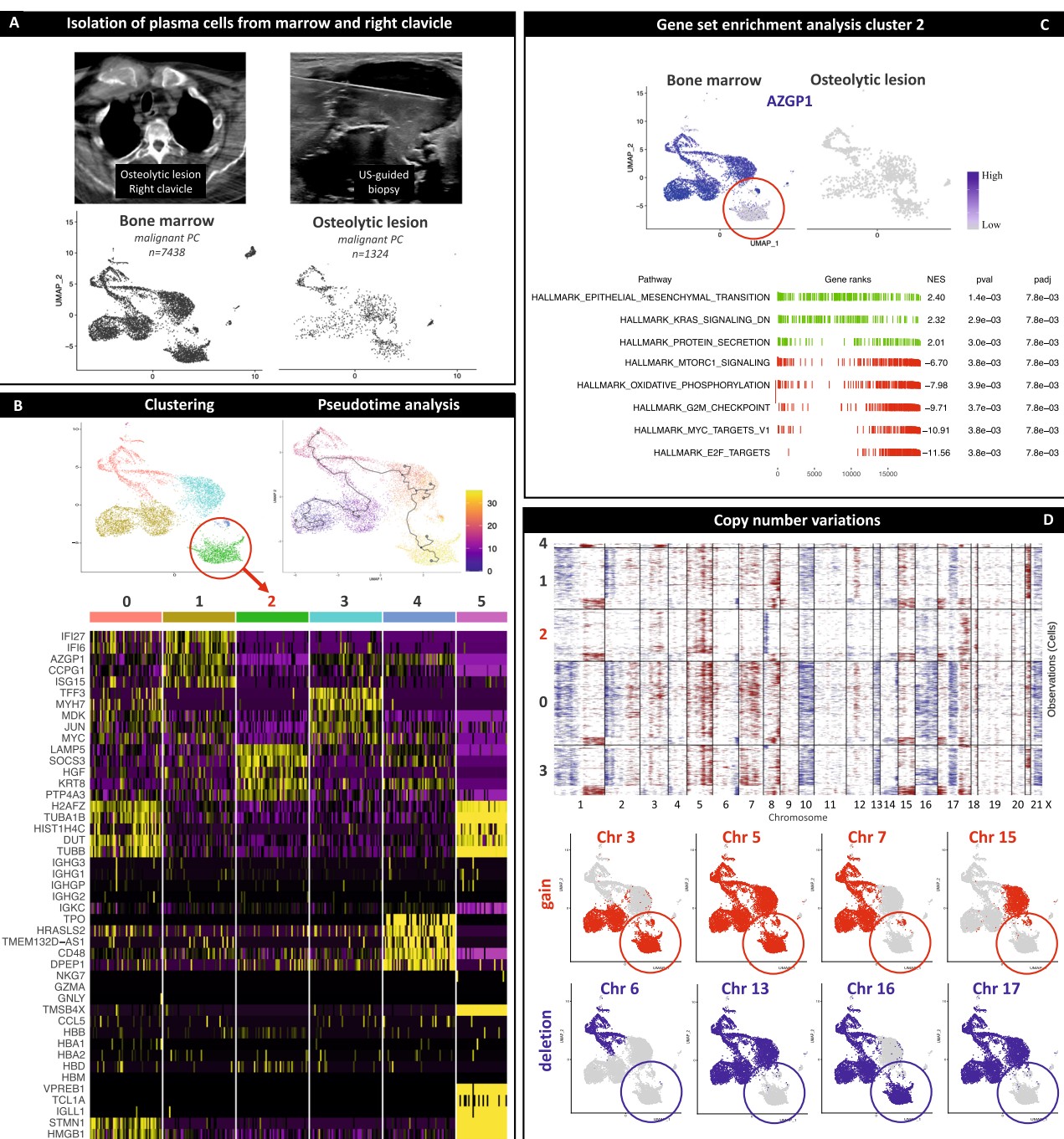

**Fig. 6 Single cell RNA sequencing links site-specific gene expression to the development of para-medullary myeloma bone disease. A** CT image of an osteolytic lesion (OL) in the right clavicle with an adjacent para-medullary tumor (left) and the respective ultrasound-guided biopsy (right) in patient RRMM01 and the corresponding uniform manifold approximation and projection (UMAP) plots split by the origin of plasma cells ((PC) lower panel). **B** Differential expression analysis showed over-expression of bone disease-associated genes (e.g., *HGF* and *LAMP5*, see heatmap) in cluster 2 that underwent the largest transcriptional change compared to the remaining malignant PC as shown by trajectory analysis. UMAPs amd heatmap show results for PC after filtering out non-PC. Relative downregulation is represented by magenta bars, upregulation by yellow bars, absent expression by black bars. **C** The majority of malignant PC in the BM showed higher expression values auf *AZGP1* compared to the OL (relative expression ranging from low in gray to high in purple). However, absent expression was detected in cluster 2 (red circle). Gene sets associated with epithelial-to-mesenchymal (EMT) transition were enriched (green bars) in the respective cluster while pathways associated with proliferation and oxidative phosphorylation were downregulated (red bars). Padj = adjusted *p*-value estimation based on an adaptive multi-level split Monte-Carlo scheme **D** inferCNV demonstrated heterogeneity on a chromosomal level with distinct CNV profiles in the detected clusters (gains in red, losses in blue). This was in contrast to findings from patients with intra-medullary disease. Upper part of the panel represents exome-wide inferred CNVs for every sequenced malignant PC (rows) and chromosome (columns). Lower part of the panel visualizes CNVs for the identified clusters and demonstrates that cluster 2 with EMT-like signatures had also a distinct CNV pattern. These findings support the preclinical hypothesis that hypoxia-driven EMT-like processes cause extra-/para-medullary spread of myeloma and that spatial heterogeneity is more significant in patients with para-/extra-medullary disease.

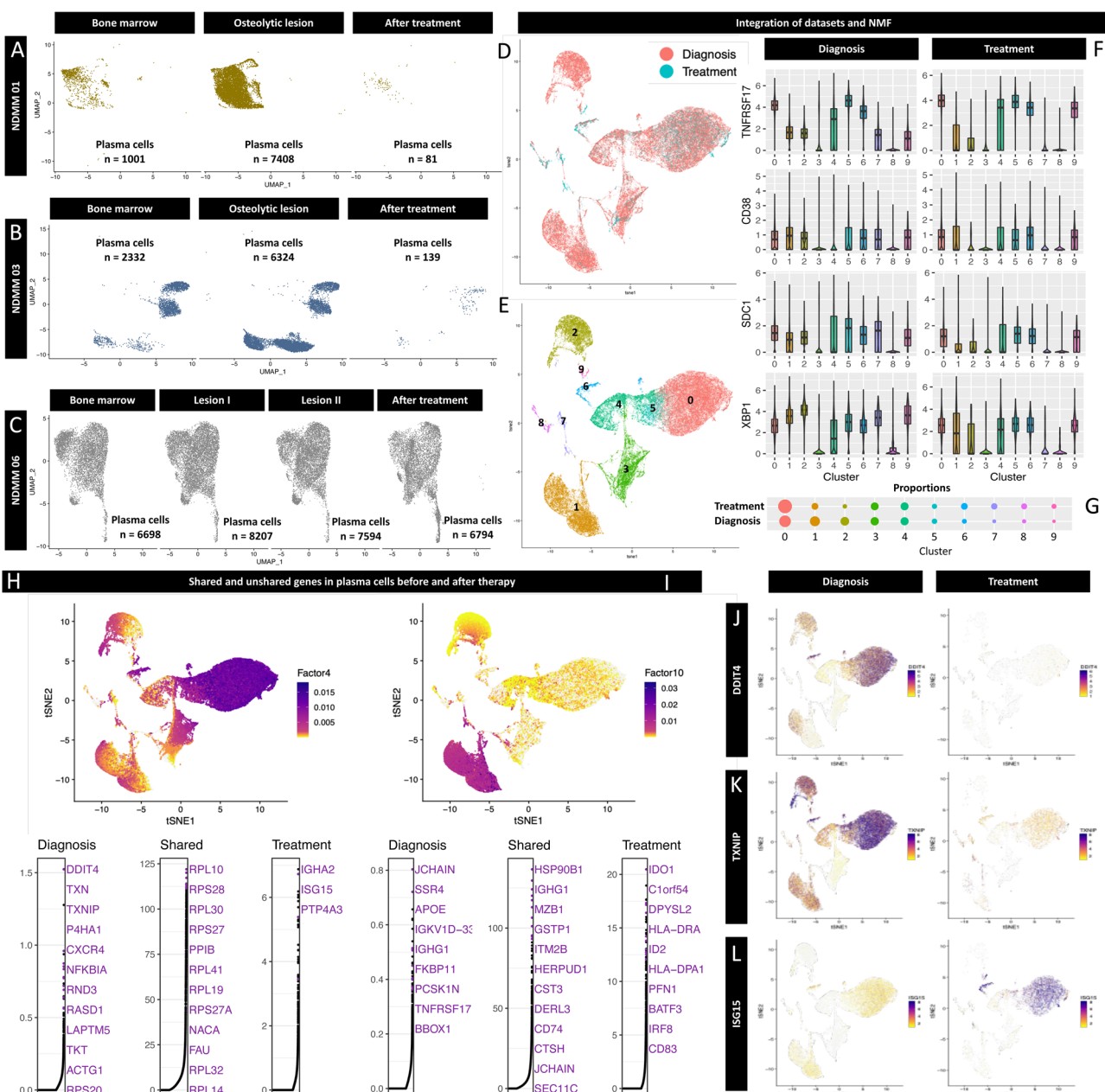

**Fig. 7 Characteriztion of measurable residual disease on a single cell level in longitudinal samples.** Uniform manifold approximation and projection (UMAP) plots and number of sequenced malignant plasma cells (PC) split by the origin of PC for patients NDMM01 **A** and NDMM03 **B** after 4 cycles lenalidomide, bortezomib and dexamethasone (RVD) as well as NDMM06 **C** after 4 cycles daratumumab-RVD. While NDMM01 and NDMM03 were in complete remission (CR) with measurable residual disease (MRD), NDMM06 was in partial remission (PR). Linked inference of Genomic Experimental Relationships (LIGER) was used for integrative non-negative matrix factorization to identify shared and unshared genes in longitudinal samples before and after treatment. Cells were visualized based on treatment situation (**D**, diagnosis (red) versus treatment (green)) and the identified clusters **E**. The majority of sequenced cells after therapy represented PC as shown by expression of common PC markers with median expression represented by horizontal lines, interquartile range by boxes and 95% confidence interval by whiskers in the respective box plots **F**. Proportions of the identified clusters before and after therapy are visualized by dotplots in **G**. Using LIGER, shared and unshared genes between diagnosis and after treatment were visualized based on the identified factors **H**, **I**. *DDIT4* **J** and *TXNIP* **K** were downregulated in the majority of PC after therapy compared to primary diagnosis. Both genes are involved in mediating cellular response to steroid exposure[34]. Besides genes encoding HLA Class II molecules, *ISG15* was significantly upregulated in PC after treatment and has been linked to carfilzomib-resistance in the past **L**[34]. Relative expression ranging from low in yellow, to high in purple.

studies in larger cohorts of homogeneously treated patients are needed to confirm these initial findings.

## Discussion

In this prospective study, we established a reproducible workflow to obtain and isolate viable PC of comparable quality from BM and paired OL. Our study links the accumulation of malignant PC with distinct transcriptomes to the development of myeloma bone disease and identifies transcriptional changes in residual disease.

scRNA-seq has been successfully utilized to study PC heterogeneity by Ledergor et al. who sequenced 24,126 single cells from random bone marrow samples of 40 individuals (11 normal

controls and 29 patients with PC disorders of which 12 were MM patients)[10]. After removal of non-PC ($n = 3,179$, 15.4%), 20,568 single PC were analyzed with the focus on inter-patient and inter-diagnostic differences. In the current study, we analyzed 148,630 PC from 24 different locations (BM and paired distant OL) in 10 individuals with MM.

We confirm that subsets of single malignant PC in individual patients can be delineated based on single cell transcriptomics and inferred CNVs. We have identified subclusters of malignant PC that are characterized by the overexpression of genes associated with proliferation and oxidative phosphorylation. Further analysis revealed that the respective clusters also show over-expression of genes that contribute to a validated GEP high-risk score (UAMS-17)[6]. Higher expression levels of two marker genes of the respective clusters were associated with adverse outcome in the CoMMpass dataset underlining the prognostic potential. Given the small number of cells compared to the entire population of malignant PC in each patient, their presence would have been missed by bulk sequencing.

Beyond the identification of inter- and intra-patient heterogeneity, we aimed at deciphering spatial heterogeneity and the underlying biology of OL. Currently, one retrospective study, examining frozen samples, used WES and GEP and demonstrated spatial genomic heterogeneity by multi-regional sequencing in more than 75% of their analyzed patients ($n = 51$)[7]. In contrast, our study demonstrated a high concordance between PC from BM and OL based on WES. This is similar to a recent WES study finding concordance in clonal somatic mutations between PC from circulating tumor cells and matched tumor biopsies from bone marrow PC[35]. Possible explanations for these differences might be the retrospective nature (analyzing previously collected and stored samples) of the Rasche et al. study and the inclusion of more patients with advanced disease (larger lesions and extra-medullary disease). This may have introduced sampling bias as there was a strong connection between spatial heterogeneity and lesion size. In the current study using prospective acquisition of fresh PC from intra-medullary locations, we demonstrated that there were greater than 80% shared mutations between both locations with the exception of patient RRMM01 with para-medullary disease and significant spatial heterogeneity. This is in agreement with the study by Rasche et al. that found limited spatial heterogeneity from WES in patients with smaller lesions[7].

Analyzing scRNA-seq data from paired OL and BM, revealed another layer of complexity and additional detail to the under-standing of PC heterogeneity. Although single PC transcriptomes from both locations were highly correlated to each other, we identified differentially expressed genes in each patient through integrated analysis of anchored datasets. Beyond genes that have been associated with myeloma bone disease (*TIMP1, HGF*[25]) and impaired Wnt-signaling (*DKK1*[24]), we identified *LAMP5* to be overexpressed in OL. Our findings connecting *LAMP5* to the development of OL is supported by a recent retrospective trial in which bulk GEP was performed on PC from patients with smoldering myeloma with and without progression during follow-up. *LAMP5* was significantly overexpressed in patients with disease progression and 8 of 10 patients progressed with new OL[36].

Additionally, *JUN/FOS*, *DUSP1*, and *HBB* were consistently downregulated in PC from OL. These genes have been correlated with extramedullary spread of MM[26]. *JUN/FOS* downregulation have been associated with malignant PC transformation[27] and progression of MM in a recent preclinical study[28]. GSEA confirmed that downregulation of genes connected to regular B-cell function were a common feature of PC from OL.

The most significant differences between OL and BM were observed in the patient with RRMM and para-medullary spread.

We provide clinical evidence for the pre-clinical hypothesis that hypoxia-driven EMT-like processes drive extramedullary spread of MM[30]. Our findings and recent preclinical studies[28] support the hypothesis that OL are derived from a common PC ancestor developing molecular features to cause myeloma bone disease. Thus, MM would behave like a solid tumor with PC metastasizing to distant locations, inducing OL.

Beyond transcriptional changes in PC from OL, we were able to detect changes upon therapy in patients with residual disease. In two MRD-positive patients after 4 cycles of RVD and one patient in PR after Daratumumab-RVD, we found an upregulation of HLA class II genes consistent with immunomodulatory effects on PC described for lenalidomide in preclinical studies[37]. Furthermore, genes associated with carfilzomib-resistance and steroid-induced apoptosis were dysregulated in residual PC. These results indicate that scRNA-seq is able to indentify and characterize residual disease. The respective changes in single PC transcriptomes may help identify new strategies to eradicate MRD in the future.

Limitations of this study include the small number of analyzed patients and the lack of preclinical validation. Therefore, our findings regarding para-medullary spread of disease and residual MM after therapy are to be interpreted with caution. Based on our current results, future analyses will follow to investigate the biological differences between para- and extramedullary disease. Establishing a translational workflow for scRNA-seq from clinical samples from different locations was one of the major goals of our study. Based on our results, future analyses will include larger numbers of patients and also patients with extramedullary disease in addition to para-medullary MM, to decipher the biological differences between both conditions.

We have shown that scRNA-seq is feasible to analyze pro-spectively transcriptional heterogeneity in fresh clinical PC samples. Site-specific accumulation of malignant PC with a distinct transcriptomic profile can be linked to the development of myeloma bone disease. We anticipate this study will con-tribute to the current understanding of MM heterogeneity and have implications for therapeutic decision-making and long term monitoring.

## Methods

**Prospective trial of imaging-guided biopsies.** This study was approved by the Roswell Park Comprehensive Cancer Center (Roswell Park) Institutional Review Board and was conducted in accordance with the Declaration of Helsinki. In April 2019 we initiated sample collection to analyze spatial and temporal evolution in newly diagnosed and relapsed MM. After written informed consent, patients underwent an imaging-guided biopsy of OL identified by PET/CT in addition to standard, diagnostic bone marrow aspirate from the iliac crest (BM). Biopsies were performed before the initiation of local or systemic therapy for newly diagnosed or relapsed patients. Eligible patients with a confirmed diagnosis of MM according to International Myeloma Working Group (IMWG) criteria were at least 18 years of age with an Eastern Cooperative Oncology Group (ECOG) performance status of 0-2 and no contraindications against general anesthesia. Key exclusion criterion was a history of other malignancy except if the patient had been symptom-free and without active therapy for at least 5 years. Patients were treated as standard of care and the procedure to obtain biopsies for this study did not affect their care plan. Patients did not receive financial compensation for participating in the study.

**Medical imaging.** All newly diagnosed MM patients underwent a PET/CT according to the current IMWG guidelines[38]. Imaging was performed on a GE Discovery ST PET/CT (GE Healthcare, Chicago, IL). Approximately 60 min after the intravenous administration of 10 mCi of 18F-fluorodeoxyglucose (FDG) a low-dose, non-contrast, diagnostic quality whole-body CT was performed for assess-ment of bone disease, attenuation correction and anatomic orientation. Axial images were reconstructed at 3.75 mm. Afterwards a PET covering the same anatomical regions was obtained and reconstructed with and without attenuation correction. Interpretation was performed according to the Interpretation criteria for FDG PET/CT in multiple myeloma (IMPeTUs)[39]. OL were characterized by the presence of circumscribed areas with bone loss. Increased tracer uptake was graded according to the 5-point Deauville scoring system[39].

**Imaging guided biopsies of osteolytic lesions**. Every patient with new OL on PET/CT was discussed in the Roswell Park Multiple Myeloma multi-disciplinary tumor board. If a patient had multiple new OL, the interventional radiologists (AB and RA) determined if a lesion was accessible for an imaging-guided biopsy with the least risk for peri-interventional complications. Lesions had to be identified on CT, showing bone destruction. OL PET-positivity was not required since some OL may not be PET-avid at primary diagnosis and false-negative PET scans can occur based on low hexokinase-2 expression[40]. Those patients with at least one potentially accessible lesion were offered to study participation. After written, informed consent for the entire study, patients were scheduled for a biopsy either under CT or fluoroscopic guidance. Patients were placed in a prone position, local anesthesia of the skin and soft tissue surrounding the OL was performed using a combination of 1% lidocaine and 0.25% preservative free bupivacaine. Under intermittent CT/fluoroscopy guidance, a 13-gauge trocar needle was advanced to the bone surface. A powered bone access system (Arrow® OnControl®, Teleflex, NC) was used to advance the needle through the outer cortex to reduce pain, procedure time and improve specimen quality[41]. After CT/fluoroscopy-guided confirmation of the correct needle placement in the OL, 10 ml bone marrow aspirate and a core biopsy were obtained of the OL. The similar process was performed on the iliac crest for the diagnostic aspirates (BM).

**Obtaining viable plasma cells from diagnostic bone marrow aspirates and osteolytic lesions**. Figure 1 summarizes the translational workflow in our prospective trial. Bone marrow aspirates from the BM and OL were collected in tubes containing Ethylenediaminetetraacetic acid (EDTA, BD Vacutainer®, BD, NJ). Plasma cells (PC) were immediately isolated using a CD138 positive selection kit according to manufacturer's instructions (EasySep™, STEMCELL Technologies, Vancouver, Canada). Cell numbers and viability of the positive and negative fraction after PC separation were checked by Trypan Blue using an automated counter (Countess™ II, Thermo Fischer, MA). PC purity was assessed by fluorescence-activated cell sorting (FACS). In one patient, RRMM01 who had an OL of the right clavicle with adjacent para-medullary disease (PMD), we sampled a soft tissue tumor that was mechanically disintegrated to bring cells into suspension.

On the same day, PCs were resuspended in RPMI 1640 containing 10% fetal bovine serum (FBS) and subjected to scRNA-seq. PCs that were not transferred for scRNA-seq were resuspended in FBS containing 10% Dimethysulfoxide (DMSO) and frozen at −80 °C.

In total, 167,453 of the initially sequenced 220,654 single cells from BM, OL and after therapy (median 7712 cells/sample) were captured after filtering out cells characterized by multiplets and high mitochondrial RNA expression that would be reflective of a high fraction of apoptotic or necrotic cells. Quality assessment revealed that 77.2% ($n = 74,922$) of cells from OL and 71.9% ($n = 73,814$) of cells from BM passed the filtering process (Table 1).

**Fluorescence-activated cell sorting (FACS)**. Immunophenotyping for PCs was performed on purified bone marrow samples from BM and OL aspirates. The following cell surface markers were used on fresh samples to identify plasma cells: CD38, CD138 and CD45[42].

**Fluorescence in situ hybridization (FISH)**. FISH analyses were performed on CD138-purified plasma cells counting at least 100 nuclei per sample and using probes for: 1q, 1p, 5q, 9 satellite III, del13q, 15 alpha satellite, del17p, t(4;14), t(11;14), t(14;16) and breakapart probes for IgH as well as MYC.

**Single cell RNA sequencing (scRNA-seq)**. Single cell gene expression libraries were generated using the 10X Genomics platform, as described previously[9]. In brief, cells were loaded into the Chromium Controller (10X Genomics, CA). Cells were partitioned into nanoliter-scale Gel Beads-in-emulsion with a single barcode per cell. After reverse transcription, the cDNA was amplified and used to generate libraries by enzymatic fragmentation, end-repair, a-tailing, adapter ligation, and PCR to add Illumina compatible sequencing adapters. The libraries were evaluated on D1000 screentape using a TapeStation 4200 (Agilent Technologies, CA), and quantitated using Kapa Biosystems qPCR quantitation kit for Illumina (Illumina Inc., Ca). They were then pooled, denatured, and diluted to 350pM with 1% PhiX control library added. The resulting pool was loaded into the appropriate NovaSeq Reagent cartridge and sequenced on a NovaSeq6000 following the manufacturer's recommended protocol (Illumina Inc., CA).

Cell Ranger (v3.1.0) was used to read alignment, filter, barcode and for UMI counting. Analyses of scRNA-seq data were performed using the Seurat R toolkit (v3.2.2) for single cell genomics[23]. The matched BM and OL samples from the same patient were merged together in the analysis. We filtered out the low quality or dying cells with more than 10% counts originating from the mitochondrial genes. The cells detected with less than 500 or more than 7,500 unique genes were also discarded to avoid empty droplet or multiplets. Data were normalized using the LogNormalization method from Seurat using a scale factor of 10,000. After feature selection and scaling of the normalized data, we performed PCA linear dimensional reduction. The first 30 PCs were used to construct the KNN graph and the Louvain algorithm was performed for clustering the cells with a resolution parameter set to 0.2. We ran the UMAP method for the non-linear dimensional reduction to visualize the dataset. Gene expression profiles were annotated with publicly available datasets (Blueprint and ENCODE) using the R package SingleR (v1.8.0)[43]. Cell cycle phases were scored using the list of cell cycle markers from Tirosh et al. that are preloaded in Seurat[44]. Trajectory inference and pseudotime calculations with Monocle 3 (v1.0.0)[45] on Seurat objects were performed using Seurat Wrappers. To characterize co-regulated gene modules across samples and patients, we used similarity weighted nonnegative embedding (SWNE) on the entire dataset[46]. Longitudinal samples from primary diagnosis and in remission were integrated and analyzed by Linked Inference of Genomic Experimental Relationship (LIGER) (v0.5.0)[31] to identify shared and unshared marker genes of malignant PC before and after therapy.

**Copy number variations from scRNA-seq**. CNVs were calculated using the inferCNV R package (v1.3.3). Count matrices and cell annotations were extracted from the Seurat S4 object for very individual patient. Reference cell clusters were determined by the annotations from SingleR. Since SingleR cannot differentiate between malignant and non-malignant PC, the latter were determined by restricted expression of heavy and light chains. CNV prediction via hidden markov model (HMM) was performed at the level of subclusters instead of the entire sample by using the random trees method and setting the p-value to 0.05. CNV predictions from HMM were visualized on UMAPs in Seurat by using the add_to_seurat command from inferCNV.

**Integrated analysis of paired samples**. To identify clusters that are present in the OL and BM and to investigate differentially expressed genes between both conditions, we performed an integrated analysis as described previously[23]. After identifying anchors between paired samples and integrating datasets for every individual patient, the standard Seurat workflow for clustering and visualization was performed. To get a broad overview on differences in gene expression between malignant OC from OL and BM, scatterplots for average gene expression were generated. Next, we identified differentially expressed genes between the two groups by using the two-sided Wilcoxon Rank Sum test. Results were visualized with the EnhancedVolcano R package (v1.12.0)[47].

**Gene set enrichment analysis**. Gene set enrichment analysis (GSEA) was performed using the FGSEA R package (v1.20.0)[48] to investigate whether scRNA-seq data can be used to classify individual patients according to the molecular classification of MM from bulk GEP[18]. The AddModuleScore() command from Seurat was used to calculate the average expression for each molecular subgroup and visualize results in a FeaturePlot.

**Whole exome sequencing (WES)**. DNA was extracted from frozen plasma cells using kits according to manufacturer's instructions (DNeasy kit, Qiagen, Hilden, Germany) for bulk WES. Oral swabs (oragene·DNA, DNA genotek, Ontario, Canada) were collected for germline controls. After quality check (Quibit Fluorometric Quantification DNA and RNA Assay kits, Thermo Fisher, MA), samples were subjected to WES. SureSelect XT Low Input Target Enrichment System (Agilent Inc, CA) was used for individual exome capturing of each DNA sample. DNA was sheared using a Covaris S220 (Covaris Inc., MA) followed by end repair, P5 adaptor ligation, and 10 cycles of PCR to complete the P7 adapter. Unique dual-indexed libraries were purified with AMPureXP beads (Beckman Coulter, CA) and validated for appropriate size on a Tapestation 4200 DNA1000 screentape (Agilent Inc., CA). The purified library was then hybridized to the SureSelectXT Human All Exon V7 Capture library (Agilent Inc., CA). Afterwards, the hybridized regions were bound to streptavidin magnetic beads and washed to remove any non-specific bound products. Eluted library underwent a second 10 cycle PCR amplification to generate enough material for sequencing. Final libraries were purified, measured by Tapestation 4200 DNA1000 screentape, and quantitated using KAPA qPCR (KAPA Biosystems, Basel, Switzerland). Individual libraries were pooled in equimolar fashion at 2 nM final concentration. Each pool was denatured and diluted to 350pM with 1% PhiX control library added. The resulting pool was then loaded into the appropriate NovaSeq Reagent cartridge and sequenced on a NovaSeq6000 following the manufacturer's recommended protocol (Illumina Inc., CA).

High quality paired-end reads passing Illumina RTA filter were initially processed against the NCBI human reference genome (GRCh37) using publicly available bioinformatic tools[49,50] and Picard (http://picard.sourceforge.net/). Putative single nucleotide variants (SNVs) and insertions and deletions (indels) were identified by running variation detection module of Bambino[51]. All putative SNVs were further filtered based on a standard set of criteria to remove the following common types of false calls: (1) the alternative allele was present in the matched normal sample and the contingency between the tumor and normal samples was not statistically significant; (2) the mutant alleles were only present in one strand and the strand bias was statistically significant; (3) the putative mutation occurred at a site with systematically dropped base quality scores; (4) the reads harboring the mutant allele were associated with poor mapping quality. Ambiguous cases were manually inspected to ensure accuracy. Putative indels were evaluated by a re-alignment process to filter out potential false calls introduced by unapparent germline events, mapping artifacts and homopolymer. The identified

somatic mutations were compared to the public human germline databases including dbSNP[52], 1000 Genomes Project[53] and the National Heart, Lung, and Blood Institute's Exome Sequencing Project to further exclude remaining germline polymorphisms. All mutations were annotated using ANNOVAR[54] with NCBI RefSeq database.

**Reporting summary**. Further information on research design is available in the Nature Research Reporting Summary linked to this article.

## Data availability

The raw scRNA-seq and WES data generated in this study have been deposited in the Sequence Read Archive (SRA) database under accession code PRJNA723584.

## Code availability

Custom code that was created for WES and scRNA-seq analyses can be requested by contacting the corresponding authors.

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

## Acknowledgements

We would like to thank our healthcare assistants Amy McCracken and Mahfoudh Saleh for their support during acquisition of the samples, preparing and caring for our patients. Special thanks to L. Shawn Matott for his continuous support regarding any problems with the High Performance Computing environment at Roswell Park. This work is supported by a grant from the German Cancer Aid covering the salary of Maximilian Merz and an Institutional Educational Research Grant from Celgene Corporation. The Genomics, Biostatistics & Bioinformatics, and Flow & Image Cytometry Shared Resources are supported by the NIH/NCI Cancer Center Support Grant P30CA16056. Hemn Mohammadpour is funded by F32 CA239356. Other support came from the Black Swan Research Initiative of the International Myeloma Foundation. Parts of this work were awarded with a Conquer Cancer Award at the 2020 ASCO meeting and presented at the ASH meeting in 2020.

## Author contributions

Conception and design: Maximilian Merz, Almuth Maria Anni Merz, Philip McCarthy and Jens Hillengass. Acquisition of data (acquired and managed patients, provided facilities, FISH, Flow cytometry, etc.): Maximilian Merz, Almuth Maria Anni Merz, Ahmed Belal, Ronald Alberico, AnneMarie W. Block, Cherie Rondeau, Kimberly Celotto, Hemn Mohammadpour, Megan Herr, Theresa Hahn, Paul K. Wallace, Joseph Tario, Jesse Luce, Sean T. Glenn, Prashant Singh, Philip L. McCarthy and Jens Hillengass. Analysis and interpretation of data (e.g., statistical analysis, biostatistics, computational analysis): Maximilian Merz, Almuth Maria Anni Merz, Jie Wang, Lei Wei, Qiang Hu, Nicholas Hutson, Mehmet Samur, Nikhil Munshi, Song Liu, Philip McCarthy and Jens Hillengass. Writing, review, and/or revision of the manuscript: All authors.

## Competing interests

P.L.M.: Advisory Board/Consulting: BlueBird Biotech, Bristol-Myers Squibb, Celgene, Fate Therapeutics, Janssen, Juno, Karyopharm, Magenta Therapeutics, Sanofi, Takeda; Honoraria: BlueBird Biotech, Bristol-Myers Squibb, Celgene, Fate Therapeutics, Janssen, Juno, Karyopharm, Magenta Therapeutics, Sanofi, Takeda. JH: Advisory Boards/ Honoraria: Adaptive, Amgen, Bristol-Myers Squibb, Celgene, GlaxoSmithKline, Janssen, Oncotracker, Oncopeptide, Skyline, Takeda. M.M.: Advisory Boards/ Honoraria: Amgen, BMS, Celgene and Takeda. Institutional Research Support: Celgene to P.L.M. and M.M. The remaining authors declare no conflict of interest.
