## [Peer review file · Nature Communications]

REVIEWER COMMENTS

Reviewer #1 (Remarks to the Author): Expert in multiple myeloma genomics and single-cell RNA-seq

In this study, the authors attempted to study multiple myeloma osteolytic lesions at genomic and transcriptomic levels. They applied scRNA-seq strategy on paired bone marrow (BM) and OL plasma cells and identified typical transcriptomic alterations at the OL sites. The paired BM and OL comparison is novel, and some findings are interesting. However, major conclusions are from limited data and findings and over-exaggerated.

Some concerns:

Page 6, related to Figure 1B: It would be important to give an overview of how cell clustering was conducted when BM and OL are combined. Do the cells from the same patient tend to cluster together or do cells from the same site tend to cluster more closely?

Page 8, related to Figure 2A and 2B: the authors pointed out a small subcluster within patients characterized by higher STMN1 expression. Are there other genes highlighted in this cluster as well? It would be good if they could do a deeper search and see if any pathways pop out. The authors also look into the CNV pattern, but the information gained is very limited – the authors could have made a better connection, for example, does the STMN1-high-population have a CNV pattern that is different from the rest of the cells?

Page 9, related to Figure 3A: have the authors checked whether the mutations are also present at the single cell level (based on snRNA data)? If there are, then it would be interesting to link mutation information at individual cell level with expression. Also, it would be good to check if any mutations may co-occur.

Page 10, related to Figure 3D: FOS and JUN are also stress markers, and HBB is a marker for erythrocytes. Therefore, it would be important to check the scRNA QC parameters and the expression of other erythrocytes markers for the two groups to see if this could be due to other effects, such as erythrocytes contamination.

Page 11, description about AZGP1: whether AZGP1 by itself is sufficient to inhibit OL is not clear in MM. So making further assumptions from this point may not be convincing. Apart from AZGP1, are there other genes that are differentiating the small cluster with the larger proportion in BM? What are they and what pathways do they contribute to?

Page 12, regarding sample collection pre/post induction: it would be good to summarize the treatment regimes in a table.

Page 14: MRD could be identified clinically, so what is the difference between clinical MRD and "scRNA-seq identifies MRD"? For the PR case alone, are these changes sufficient to conclude resistance? It is difficult to draw conclusions from limited longitudinal samples; therefore, the statement is exaggerated.

Page 40 : Fig 3A: Using different grey level for different locations could be quite confusing, especially for cases where one of the locations is the dominant (e.g. NDMM06 and NDMM05).

Reviewer #2 (Remarks to the Author): Expert in cancer genomics and single-cell RNA-seq

In “Deciphering spatial genomic heterogeneity at a single cell resolution in multiple myeloma”, the authors analysed malignant plasma cells from paired bone marrow aspirates and osteolytic lesions using single-cell RNA-seq in order to investigate spatial heterogeneity of multiple myeloma. Their results showed that the single-cell transcriptomics from the two locations were correlated and their genetic profiles similar, however there were several differentially expressed genes potentially associated with myeloma bone disease and development of OL.

Unfortunately, the manuscript does not provide significant new insights into the spatial heterogeneity and progression of multiple myeloma and the underlying biology of osteolytic lesions. As such, several points should be improved before this manuscript can be accepted:

- Malignant cells usually cluster per patient due to genetic alterations, and therefore the marker genes per each patient are not too informative in terms of shared gene programs/modules. It would be good to further investigate whether there are co-regulated gene modules across tumours, or within a molecular subtype, using Non-negative matrix factorisation for example.
- The molecular subtypes of MM should be briefly explained for non-experts (page 6).
- It was not clear whether both BM and OL cells were presented in the UMAPs in figure 2, because the authors mention integrating BM and OL cells per patient later, in figure 3. Were those clusters identified in both BM and OL? Were for example the STMN1 TUBA1B subpopulations more enriched in BM vs OL? What are the profiles of other clusters? Are they similar across patients or within a molecular subtype, or was only the STMN1 TUBA1B cluster common across patients?
- Figure 2C. Are these genetically identified subclones also transcriptionally different? It would be good to see those subclones on UMAP plots to see whether they overlap the transcriptional clusters.
- The MRD and therapy part was very brief, the authors only mention a few genes up or downregulated without any new insights or further experimental validation.
- Panel D from Figure 4: Is this trajectory from the BM cells only? Do you see a trajectory going from the AZGP1 negative BM cells to the OL cells? Only some of the OL cells cluster with those AZGP1- BM cells, many OL cells cluster with the other AZGP1 cells, how do you explain this? Are there any interesting genes upregulated along the trajectory, potential transcription factors driving this transition?
- There was no further experimental validation of the potentially interesting subpopulations (STMN1 TUBA1B subtype) or the DE genes between OL and BM lesions or changes upon therapy, for example in situ validation or by isolating those subpopulations. While it is appreciated that STMN1 and TUBA1B are associated with shorter survival, there were also no further functional analyses.

Reviewer #3 (Remarks to the Author): Expert in multiple myeloma genomics and pathogenesis

Merz and colleagues describe scRNA-Seq from paired random bone marrow and osteolytic lesion sites in 10 patients with myeloma. They also describe longitudinal scRNA-Seq from 3 patients. The manuscript is beautifully written and very easy to follow. They do not waste the reader's time by extensively recapitulating findings from bulk RNA-Seq (though they do provide brief analyses that give us confidence in the data), but instead focus on examining the intra-patient comparisons which are enabled by this technology. Key findings are the differential expression of genes between the BM and OL sites within patients and the differential expression of genes between diagnosis samples and MRD samples. I believe that this paper adds significantly to the myeloma scRNA-Seq literature.

I have only minor comments:

The survival comparisons in fig. S2 are for allcomers. CoMMpass represents very heterogeneously treated patients and it may be that this is affecting the survival analysis. There should be sufficient patient numbers in CoMMpass to also look at broad treatment groups (say IMiD versus PI versus joint IMiD/PI therapy). It may be that expression of these genes is relevant for particular treatments.

Although I think I understand, it is not explained entirely clearly why, for example, DKK1 differs between OL and BM after anchoring (page 10) but not in the earlier analysis (page 6). Although the anchoring procedure is described in the reference and methods, a line or two of explanation within the results section would be helpful to understand the discrepancy.

In a similar vein, can we be reasonably confident that these differences between OL and BM would not have been discovered by bulk sequencing of samples from both sites followed by some sort of paired analysis (e.g. Wilcoxon signed rank test)? I.e. it would be good to fully justify the use of scRNA-Seq for these key differential expression findings.

Page 11: "AZGP1 is a known tumor suppressor gene causing mesenchymal-to-epithelial transition and inhibiting invasion and metastasis". Should this not read "AZGP1 is a known tumor suppressor gene whose downregulation causes mesenchymal-to-epithelial transition..."

My understanding of the differential expression of AZGP1 is was that it was seen in a single sample. If this is correct, the authors should caution the reader against over-interpretation of the AZGP1 data (although the pseudotime analysis is extremely nice!).

Reviewer #4 (Remarks to the Author): Expert in osteolytic lesion detection and haematologic malignancies

The authors reported comparative single-cell RNA sequencing analysis of bone marrow plasma cells from random bone marrow aspirate and image-guided biopsy of the osteolytic lesions in patients with newly diagnosed and relapsed myeloma.

Clonal heterogeneity in the same patient has well been recognized in myeloma. In addition, the spatial heterogeneity between the focal lesions and traditional iliac crest biopsies has been documented in large study by the UAMS group using numerical or structural chromosomal aberrations, short insertions or deletions or single nucleotide variants. This study uses single cell RNA sequencing, a much more sensitive technology that allows a deeper dive the single-cell transcriptomes.

The study uses small sample size of 7 newly diagnosed and 3 relapsed cases and therefore the results are thought-provoking and represent a feasibility study and a proof of concept. Clinical correlation is limited and require a larger study. Paired sample analysis of the residual tumor cells post-treatment is interesting but whether these cells represent a driver clone for relapse is not clear. In addition, whether the specific genes preferentially expressed in the OL PCs compared to BM PCs represent a subclone or expression alteration due to factors in the tumor microenvironment remain unknown, and perhaps should be discussed. The intra and inter- patient heterogeneity are as to be expected but intrigue data is in the much different single cell transcriptomics identified when WES only demonstrate limited differentially expressed genes.

Based on the objective of this report being the technical feasibility, more details should be provided on the chosen patients and the biopsies. Clinical information regarding the extent of bone involvement is needed. The number and size of the lesions were not described. One lytic lesion biopsy was obtained in each patient (except for one case where 2 sites were chosen) at the same time as the bone marrow aspirates. To fully understand how representative the one site may be of all other lytic lesions, such description would be needed.

The technical aspect of this study is of interest as their established method may inform other investigators. The authors should be commended for the concurrent biopsies and processing, to avoid the interference of processing artifact. While sample size determination for scRNAseq continues to be a controversial topic and standardized methodology has not been determined, the number of PC examined in this study is in line with others that have used the similar technique for study of drug resistance mechanism or progression from pre-myeloma conditions.

Please further specify whether there were cases that have been excluded due to inadequate PCs. Since PET/CT was used to identify these lesions, please also consider adding the information regarding FDG uptake of the chosen lesion. The decision for which lesion to choose was done by the multidisciplinary tumor board. Please discuss the general factors included in the decision on the biopsy sites.

PC viability from both biopsies was excellent and all cases had enough number of PCs that passed QC. Aspiration of lytic lesions is typically done by fine needle aspiration using a small-gauge needle compared to the bone marrow aspiration needles. However, the percent PC in the focal lesion is typically high, usually represent clusters of PCs. Was there an immediate pathologic evaluation by cytopathology to assure an adequate number of cells were obtained?

Overall, the work is innovative and now further highlights the inter-lesion heterogeneity and incorporation of concurrent biopsy from multiple sites pre- and post-treatment for targeted and personalized therapy in this incurable disease.

Reviewer #1 (Remarks to the Author): Expert in multiple myeloma genomics and single-cell RNA-seq

In this study, the authors attempted to study multiple myeloma osteolytic lesions at genomic and transcriptomic levels. They applied scRNA-seq strategy on paired bone marrow (BM) and OL plasma cells and identified typical transcriptomic alterations at the OL sites. The paired BM and OL comparison is novel, and some findings are interesting. However, major conclusions are from limited data and findings and over-exaggerated.

Some concerns

Page 6, related to Figure 1B: It would be important to give an overview of how cell clustering was conducted when BM and OL are combined. Do the cells from the same patient tend to cluster together or do cells from the same site tend to cluster more closely?

We thank the reviewer for this comment. We conducted further analyses and have provided a new figure in the manuscript (Figure 2). To analyze whether clustering of cells was predominantly driven by spatial heterogeneity or inter-patient differences, we repeated downstream analyses according to the reviewer's comment. As shown in the figure below (page 2 of this document), cells from individual patients clustered more closely together rather than samples from same locations in different patients. Regions where cells from different individuals clustered together were populated by non-plasma bone marrow cells (encircled red in figure 2D and E). This information about the relationship between inter-patient and spatial heterogeneity is important for the entire analysis. Figure 2 contains this information (described on pages 6 and 33 of the revised manuscript).

To summarize, inter-patient heterogeneity and not spatial heterogeneity determined clustering of cells in the entire dataset. We again thank the reviewer for this important comment.

Results, Page 6:

*Clustering of 148,746 single cells from BM and OL (median 7712 cells/sample) created a map of distinct populations based on transcriptomes from individual patients and locations (**Figure 2A and B**). Cells from individual patients clustered together in both, BM and OL (**Figure 2C**). This was also observed when merging cells from both locations (**Figure 2D**) which demonstrated that inter-patient heterogeneity outweighed intra-patient spatial heterogeneity and determined clustering patterns of malignant PC. Clusters with overlapping cells from different patients were later identified as few contaminating, non-PC (**Figure 2E**).*

Page 8, related to Figure 2A and 2B: the authors pointed out a small subcluster within patients characterized by higher STMN1 expression. Are there other genes highlighted in this cluster as well? It would be good if they could do a deeper search and see if any pathways pop out. The authors also look into the CNV pattern, but the information gained is very limited – the authors could have made a better connection, for example, does the STMN1-high-population have a CNV pattern that is different from the rest of the cells?

Thank you for this important comment. To gain further insight into the pathways of the sub-populations, we repeated the analyses as per the reviewer's suggestion.

First, we repeated analyses for each individual patient to characterize differentially expressed genes in clusters characterized by STMN1 expression. The figure below (Page 4 of this document) shows the results for one representative patient (RRMM03). This is now included in the manuscript (Page 8; 33 and 34, Panels A-D in Figure 4).

There was strong overlap between differentially expressed genes in the STMN1-positive clusters among all patients. We provide the entire list of genes identified by this analysis as supplemental table and show the top 5 up-regulated genes in the respective cluster compared to other clusters in the figure (Figure 4B).

To gain further molecular insights into the respective malignant plasma cells, we performed a gene set enrichment analysis on each patient sample (e.g. RRMM03 in figure 4C) to detect pathways significantly up- or down-regulated in the respective clusters compared to the other malignant cells. Using the hallmark gene set collection from the Molecular Signature Database, we identified gene sets associated with cell division, mitotic activity and oxidative phosphorylation to be enriched in the respective STMN1-pos clusters in all patients.

We furthermore inferred CNVs for every individual cluster as proposed by the reviewer (example for RRMM03 in figure 4D). However, the respective STM1-positive clusters did not show different CNVs compared to the majority of malignant plasma cells.

Based on the reviewer's comment and the comments from reviewers 2 and 3 on the impact of the AZGP1-negative cluster on para-medullary spread of MM, we also connected inferred CNVs to detected clusters (Pages 11 of this document).

A Analysis of individual patients

B Differentially expressed genes in STMN1-pos cluster

C Gene set enrichment analysis STMN1-pos cluster

D Copy number variations

Based on comment 3 from reviewer #2, we determined the representation of STMN1-positive plasma cells in bone marrow/osteolytic lesions (see panels E-J of the new figure 4 on the next page). The numbers of cells expressing STMN1 were calculated using the function published recently on the Satija's lab Github (<https://github.com/satijalab/seurat/issues/371>). Except for patient RRMM01 with a para-medullary lesion, no differences in STMN1-expressing cells were found between bone marrow and osteolytic lesion (Bar plot in panel 4G, page 6 of this document and page 8, 9, 34 and 35 of the manuscript).

Based on the additional analyses proposed by the reviewer, we hypothesized that STMN1-positive plasma cells are actively proliferating. We next used the CellCycleScoring() command in Seurat with the preloaded markers to assign and visualize cell cycle stages (UMAP in panel 4H on page 6 of this document and page 8, 34 and 35 of the manuscript). This confirmed that STMN1-positive cells are proliferative as these cells were more in G2M and S-Phase compared to the remaining malignant plasma cells in G1-Phase (Bar plot in panel 4I on page of this document and page 8 and 34 and 35 of the manuscript).

Proliferation has been identified as a central prognostic marker in multiple myeloma by bulk gene expression profiling (GEP). The percentage of proliferating plasma cells is inversely correlated to outcome. As described in the initial version of the manuscript, the IFM GEP score includes genes associated with cell division such as STMN1. To validate the prognostic significance of the identified subclusters, we calculated the UAMS17-score for each individual cell using the AddModuleScore() command (UMAP in panel 4J on page 6 of this document and page 8, 9, 34 and 35 of the manuscript).

In contrast to the IFM score consisting of genes associated with cell division and chromosomal instability, the 17 genes of the UAMS17 score map predominantly to chromosome 1. Higher UAMS17 scores were seen in STMN1-positive malignant plasma cells (Figure 4J), underlining the significance of this subcluster. Of note, the highest percentage of STMN1-/UAMS17-positive cells were found in patient RRMM01 with chromosome 1 aberrations as confirmed by inferCNV and FISH (Table 1).

In addition to the new figure 4, the sections on intra-patient heterogeneity in the results section on pages 8 and 9 as well as the discussion on page 15 have been rewritten.

As per the reviewer's suggestion, we further characterized the identified subclusters. We show that scRNA-seq enables detection of actively proliferating plasma cells, which is important since proliferation is a prognostic aspect of myeloma. The clusters have different risk scores according to the UAMS17 scoring system and are not enriched in specific locations (bone marrow/lesion) except for a single patient with para-medullary disease.

E All samples

F STMN1 positive malignant plasma cells

G Proportion of positive cells in marrow and lesion

H Cell cycle scores

I Cell cycle score proportions in marrow and lesion

J STMN1 and UAMS-17 score

Page 9, related to Figure 3A: have the authors checked whether the mutations are also present at the single cell level (based on snRNA data)? If there are, then it would be interesting to link mutation information at individual cell level with expression. Also, it would be good to check if any mutations may co-occur.

The reviewer raised a very important point. We checked whether genes that were downregulated in osteolytic lesions were affected by mutations. No mutations in WES were found in the respective genes.

We are not able to check if WES mutations are present in scRNA-seq data due to the platform limitation. As we use the 10x genomics Chromium Single Cell 3' Reagent Kits (v3 chemistry) in our current scRNA-seq study. Therefore, cDNA covering limited numbers of bps of the respective gene are generated and no further information on upstream mutations can be gathered from the current data.

Page 10, related to Figure 3D: FOS and JUN are also stress markers, and HBB is a marker for erythrocytes. Therefore, it would be important to check the scRNA QC parameters and the expression of other erythrocytes markers for the two groups to see if this could be due to other effects, such as erythrocytes contamination.

The reviewer points out the importance of sample comparability with regards to plasma cell purity, viability and quality between osteolytic lesions and bone marrow. To ensure sample viability and purity, we implemented a translational workflow with quality assessments at pre-analytical stages and downstream analysis. We have addressed this in the modified Figure 1 (Page 10 of this document and page 6, 19, 20 and 33 of the manuscript).

In step one, bone marrow aspirates and biopsies were acquired from the iliac crest and corresponding osteolytic lesions. Aspirates for diagnostic purposes were evaluated with multi-color flow cytometry (1B), Wright-Giemsa staining, metaphase karyotyping and interphase FISH. Trephine biopsies were assessed by Hematoxylin/Eosin and immunohistochemistry staining (1C) in the Roswell Park Pathology Department to confirm the diagnosis of multiple myeloma.

For the experimental assays, 10 ml aspirates were collected in EDTA tubes and transferred to the research lab. Plasma cells were isolated using CD138-labelled magnetic beads (1A). This process included application of a red cell lysis buffer according to manufacturer's protocols (EasySep, Stemcell Technologies) to prevent erythrocyte contamination. The magnetic bead sorting was accompanied by 4 washing steps.

Plasma cell purity and viability after sorting was checked by flow cytometry (2) before single cell RNA sequencing (3). The median plasma cell purity and viability after magnetic bead sorting was 96% and 92%, respectively (Table 1).

After alignment of single cell data in CellRanger, count matrices were imported into Seurat and QC parameters were assessed (4). Empty droplets, multiplets or droplets containing cells undergoing apoptosis were excluded by filtering out entries with less

than 500 or more than 7500 genes as well as cells with more than 10% of genes mapped to mitochondrial genes. In total, 77.2% of cells from osteolytic lesions and 71.9% cells from bone marrow passed QC (5), which underlines sample comparability. The numbers of cells passing QC are comparable to the first scSeq study published by Ledergor et al in Nature Medicine in 2018. After clustering of cells (6), cell type annotation was performed with SingleR to identify contaminating, non-plasma cells (7). In accordance with flow cytometry analysis after magnetic bead sorting, 94.8% of cells from the osteolytic lesion and 95.7% from bone marrow were identified as plasma cells. For further analyses, the non-plasma cells were removed *in silico*. In total, 388 removed cells belonged to the erythroid lineage. This underlines the low percentage of contamination by erythrocytes. No further genes associated with erythroid lineage - like carbonic anhydrase II – were differentially expressed. Furthermore, comparison between malignant plasma cells from osteolytic lesions and bone marrow was performed after delineating malignant from non-malignant plasma cells based on light chain restriction (10).

Based on steps 1 through 10, we ensured high purity, quality and comparability of samples from osteolytic lesions and bone marrow. The new Figure 1 illustrates the pre-analytical and analytical steps (Pages 6, 19, 20 and 33 of the manuscript).

Page 11, description about AZGP1: whether AZGP1 by itself is sufficient to inhibit OL is not clear in MM. So making further assumptions from this point may not be convincing. Apart from AZGP1, are there other genes that are differentiating the small cluster with the larger proportion in BM? What are they and what pathways do they contribute to?

Thank you for the observation. Reviewers 2 and 3 also inquired about the AZGP1 association and pathway analysis. We have repeated the entire analyses and added more information:

After identification of malignant plasma cells from both locations (Page 11 of this document: UMAPs in Figure 6A, page 12, 36 and 37 of the manuscript), we performed clustering, differential expression and pseudotime analyses to characterize cluster 2 in the bone marrow that resembled malignant plasma cells.

We added heatmaps for the top 5 up- and down-regulated genes in the respective cluster compared to the remaining bone marrow plasma cells (Page 11 of this document, Figure 6B and pages 12, 36 and 37 of the manuscript). In addition to the down-regulation of AZGP1, we show in line with our integrated comparison of plasma cells from both locations in the entire cohort, a down-regulation of JUN as well as up-regulation of LAMP5 and HGF as well as SOC53, KRT8 and PTP4A3.

We then undertook a gene set enrichment analysis for cluster 2 using the hallmark gene set curated in MSigDB (Page 11 of this document, Figure 6C and pages 12, 36 and 37 of the manuscript). Remarkably, the gene set for epithelial mesenchymal transition (EMT); often observed in metastasizing solid tumor cells, was enriched in cluster 2 compared to the other malignant bone marrow plasma cell clusters. In accordance with other pre-clinical studies (Roccaro AM et al, Cell Rep, 2015, Azab et al., Blood, 2012), we furthermore demonstrated that pathways associated with cell division, proliferation and oxidative phosphorylation were down-regulated in the respective cluster (Page 11 of this document, Figure 6C and pages 12, 36 and 37 of the manuscript).

We provide novel evidence of EMT-like changes in plasma cells from a para-medullary tumor, accompanied by decreased mitotic activity, which has been described to contribute to chemo-resistance in pre-clinical studies (e.g. Roccaro et al, Cell Rep, 2015, Azab et al., Blood, 2012). This may be one of the explanations for the poor outcome in patients with para-/extra-medullary disease.

As per the second comment from reviewer 1 and comment 4 from reviewer 2, we mapped the inferred CNVs to the identified clusters in the patient with para-medullary disease. We demonstrate that these respective clusters have different CNVs, emphasizing the heterogeneity of bone marrow plasma cells (Page 11 of this document, Figure 6D and pages 12, 36 and 37 of the manuscript).

Thank you for this important comment that let to further insightful analyses and the respective findings.

Page 12, regarding sample collection pre/post induction: it would be good to summarize the treatment regimes in a table.

We have added the respective table as supplemental information

Patient	Regimen	Lenalidomide	Bortezomib	Dexamethasone	Daratumumab	cycles
NDMM01	RVD	25 mg/d PO d1-14	1.3 mg/m ² SC d1, 4, 8, 11	20 mg/d PO d1-2, 4-5, 8-9, 11-12	-	4 21d cycles
NDMM03	RVD				-	
NDMM06	Dara-RVD				16 mg/kg IV, d1,8,15	

Page 14: MRD could be identified clinically, so what is the difference between clinical MRD and "scRNA-seq identifies MRD"? For the PR case alone, are these changes sufficient to conclude resistance? It is difficult to draw conclusions from limited longitudinal samples; therefore, the statement is exaggerated.

We agree that the sub-heading is misleading and the statement needs to be toned down based on the few paired samples before and after treatment. To objectify our results further, we performed in agreement with comment 1 from reviewer 2 non-negative matrix factorization using the LIGER R package after integrating the entire dataset pre- and post-induction treatment. We identified shared and un-shared genes between both conditions (baseline and after treatment) that might explain why residual cells survived induction treatment and to what extent the respective agents changed single plasma cell gene expression. Nevertheless, we acknowledge the limitation due to small sample sizes and underlined that this is a proof of principle for future analyses of larger datasets.

We changed the wording in the results section on longitudinal samples on pages 13 and 14, included a new figure 7 and include the following statements in the results section as well as discussion to comment the limitations due to sample size:

Results, page 14:

However, our observations were based on only three patients showing feasibility of this methodology to characterize residual cells. Future longitudinal studies in larger cohorts of homogeneously treated patients are needed to confirm these initial findings.

Discussion, page 17:

Limitations of this study include the small number of analyzed patients and the lack of preclinical validation. Therefore, our findings regarding para-medullary spread of disease and residual MM after therapy are to be interpreted with caution.

Page 40 : Fig 3A: Using different grey level for different locations could be quite confusing, especially for cases where one of the locations is the dominant (e.g. NDMM06 and NDMM05).

We increased the size of panel A showing results from WES in the respective figure 3 to increase legibility.

Reviewer #2 (Remarks to the Author): Expert in cancer genomics and single-cell RNA-seq

In “Deciphering spatial genomic heterogeneity at a single cell resolution in multiple myeloma”, the authors analysed malignant plasma cells from paired bone marrow aspirates and osteolytic lesions using single-cell RNA-seq in order to investigate spatial heterogeneity of multiple myeloma. Their results showed that the single-cell transcriptomics from the two locations were correlated and their genetic profiles similar, however there were several differentially expressed genes potentially associated with myeloma bone disease and development of OL.

Unfortunately, the manuscript does not provide significant new insights into the spatial heterogeneity and progression of multiple myeloma and the underlying biology of osteolytic lesions. As such, several points should be improved before this manuscript can be accepted:

- Malignant cells usually cluster per patient due to genetic alterations, and therefore the marker genes per each patient are not too informative in terms of shared gene programs/modules. It would be good to further investigate whether there are co-regulated gene modules across tumours, or within a molecular subtype, using Non-negative matrix factorisation for example.

We thank the reviewer for this very important comment that lead to the following analyses: After characterizing inter-patient heterogeneity and proofing applicability of established molecular subtypes from GEP to scRNA-seq data as shown in figure 3, we aimed at identifying gene modules across tumors from different patients as proposed by the reviewer. For this purpose, we used Similarity Weighted Nonnegative Embedding (SWNE) published by Wu et al. in Cell Systems, 2018 (<https://github.com/yanwu2014/swne>). SWNE uses nonnegative matrix factorization (NMF) to decompose the expression matrix into biologically relevant factors. Factors and cells are plotted onto a two-dimensional space together with selected genes. A similarity matrix is used to smooth cell, factor and gene embeddings to ensure that cells that are close in the high-dimensional space are also close in the 2D visualization. In our case, we choose heavy and light chain genes to be embedded because of their dominant expression in malignant plasma cells. As expected, malignant plasma cells from all analyzed patients clustered closely together (see new supplemental figure 1A on next page). Among the identified factors, factor_5 had the strongest pulling effect on cells. In accordance with our analyses using Seurat for plasma cells from individual patients, genes associated with proliferation (STMN1, TUBA1B, HMGB2 and H2AFZ) were the top genes contributing to factor_5 (see panel B). Therefore, NMF confirmed our findings from separate analyses of individual patients.

Remarkably, STMN1 has been identified by the group from Ido Amit in a recent single cell RNA-seq study to promote relapse and resistance to intensive treatment with Daratumumab-KRD. The respective study was published after our manuscript was submitted. Therefore, we include this excellent work now in our manuscript since it also supports our hypothesis that scRNA-seq identifies clinically relevant subclusters (see Results, page 8).

The SWNE NMF analysis is now included as supplemental figure 1.

A Nonnegative matrix factorization using similarity weighted nonnegative embedding (SWNE)

Nonnegative matrix factorization

Plotting malignant plasma cells from all patients

B Visualization of top 4 genes contributing to factor_5

Secondly, we integrated pre-/post datasets and used LIGER (<https://github.com/welch-lab/liger>) to perform NMF on our pre-/post-treatment datasets from three patients (see new Figure 7 panels A-C on next page) and to identify shared and dataset-specific markers. We visualized clusters by condition (Diagnosis vs. after treatment, (Figure 7D)). To confirm that the identified clusters (Figure 7E) were truly plasma cells, we analyzed expression of common plasma cell markers (BCMA/TNFRSF17, CD38, CD138, XBP1 in (F)). Analysis of shared and unique genes between conditions confirmed that genes that have been associated with Carfilzomib-resistance (ISG15) and Steroid-/ bortezomib refractoriness (TXNIP, DDIT4) were downregulated. Separate visualization for individual genes (J, K and L) revealed that cells from different patients showed the same pattern of up- / down-regulation of the respective genes. Here, NMF allowed to identify genes that are differentially expressed among cells from different patients that were treated with comparable regimen but achieved different depths of remission.

This underlines the hypothesis that scRNA-seq is capable to further characterize MRD in patients with MM. In line with reviewer 1, we mention now that our findings proof feasibility of this translational approach. Nevertheless, we are limited by the small number of pre-/post-samples.

Accordingly, we include now figure 7 with the results from NMF on pre-/post-treatment samples using LIGER and changed the paragraph in the results section on pages 13 and 14.

We thank the reviewer for this important comment that led to the described analyses.

- The molecular subtypes of MM should be briefly explained for non-experts (page 6).

We have added a paragraph on the explanation of different MM-subtypes defined by bulk GEP in the respective section

Results, page 7:

Since we observed vast inter-patient heterogeneity, we performed a gene set enrichment analysis (GSEA) using the curated MM subtype gene sets from the Molecular Signature Database (MSigDB) to assess whether our findings are consistent with the molecular classification of MM that was established with GEP. In agreement with the single gene analysis, gene-set analysis showed that patients could be grouped according to the molecular classification of MM. The respective classification differentiates 7 MM subtypes influenced by the presence of genetic lesions such as translocations and a hyperdiploid karyotype as well as low-incidence of bone disease or increased expression of proliferation-associated genes. Beyond the genetic and phenotypic differences between the groups, the molecular classification is also of prognostic significance with adverse outcome for patients in the proliferation (PR) group.

- It was not clear whether both BM and OL cells were presented in the UMAPs in figure 2, because the authors mention integrating BM and OL cells per patient later, in figure 3. Were those clusters identified in both BM and OL? Were for example the STMN1 TUBA1B subpopulations more enriched in BM vs OL? What are the profiles of other clusters? Are they similar across patients or within a molecular subtype, or was only the STMN1 TUBA1B cluster common across patients?

This is a very good question and in line with the comments from reviewer 1.

We have added a detailed description of the merging process for samples from different patients and different locations in the new figure 2. By addressing this important comment from the reviewer, it is now easier to understand that cells clustered together on a per-patient basis rather than based on original location. This means that inter-patient heterogeneity was more significant than intra-patient spatial heterogeneity. Interestingly, as depicted in the respective new figure, the only clusters with overlapping cells from different patients were identified as few contaminating, non-plasma cells.

We added a corresponding passage in the results section in addition to the new figure 2:

Results, Page 6:

Clustering of 148,746 single cells from BM and OL (median 7712 cells/sample) created a map of distinct populations based on transcriptomes from individual patients and locations (Figure 2A and B). Cells from individual patients clustered together in both, BM and OL (Figure 2C). This was also observed when merging cells from both locations (Figure 2D) which demonstrated that inter-patient heterogeneity outweighed intra-patient spatial heterogeneity and determined clustering patterns of malignant PC. Clusters with overlapping cells from different patients were later identified as few contaminating, non-PC (Figure 2E).

The second part of this comment addresses the important question regarding distribution of STMN1/TUBA1B-pos clusters among patients as well as locations and is in line with comments from reviewer 1. In a first step, we repeated the analysis for every individual patient to identify other up-/down-regulated genes in STMN1/TUBA1B-pos clusters (see new Figure 4A and B). As described above, gene set enrichment analysis revealed in all patients that pathways associated with cell division, proliferation and mitotic activity were enriched in the respective clusters (see new Figure 4C). This finding was confirmed by NMF using SWNE – as described above – that demonstrated relative overexpression of STMN1, TUBA1B, HMGB2 and H2AFZ in cells clustering to the factor with the strongest pulling effect on the entire plasma cell population (see new supplemental figure 1).

After showing that the respective cells were identified in all analyzed patients, we aimed at addressing the reviewer's important question regarding the spatial distribution of these clusters. To quantify the number of cells expressing STMN1. We used code that was recently published on Github (<https://github.com/satijalab/seurat/issues/371>). We found no significant differences regarding STMN1-positive cells between both locations in the majority of patients. The only differences were assessed in patient RRMM01 with para-medullary spread of the disease in the right clavicle (see new Figure 4G). The respective finding lead to further analyses, also regarding the connection between scRNA-seq clusters and CNVs as described in the answer to the next question.

The important information on spatial distribution of STMN1-pos cells is now included in the new figure 4 and the corresponding passage in the results section was changed on page 8.

- Figure 2C. Are these genetically identified subclones also transcriptionally different? It would be good to see those subclones on UMAP plots to see whether they overlap the transcriptional clusters.

While we did not identify certain CNV patterns that were specific for the STMN1-pos clusters, we repeated the analysis for RRMM01 with marked spatial heterogeneity also from WES. In accordance with the reviewer's suggestion, we used the `add_to_seurat()` command from `inferCNV` to add the information back to the initial Seurat object and visualize CNVs for each cluster with UMAPs. Remarkably, this revealed that the subclone residing in bone marrow that might have given rise to the osteolytic lesion was not only characterized by enrichment of an EMT gene set and down-regulation of genes associated with cell division, but also a distinct CNV-pattern that was different from the remaining malignant plasma cells. We thank the reviewer for this suggestion that added another level of granularity to our analysis.

The new analyses are presented in Figure 6 and we changed the paragraph in the results section on page 12.

- The MRD and therapy part was very brief, the authors only mention a few genes up or downregulated without any new insights or further experimental validation.

In line with the comment from reviewer 1 regarding this section, we repeated the analysis of our pre-/post-therapy samples to objectify our findings. By using NMF, we identified differentially expressed genes that are (un-)shared between patients and conditions (prior vs. after treatment) as described above.

Unfortunately, for this manuscript, we were not able to perform *in vitro* validation of our findings since the scope of our study was to focus on patient samples and establish a clinically applicable workflow. We included a comment of limitation in the results and discussion section:

Results, page 14:

However, our observations were based on only three patients showing feasibility of this methodology to characterize residual cells. Future longitudinal studies in larger cohorts of homogeneously treated patients are needed to confirm these initial findings.

Discussion, page 17:

Limitations of this study include the small number of analyzed patients and the lack of preclinical validation. Therefore, our findings regarding para-medullary spread of disease and residual MM after therapy are to be interpreted with caution.

- Panel D from Figure 4: Is this trajectory from the BM cells only? Do you see a trajectory going from the AZGP1 negative BM cells to the OL cells? Only some of the OL cells cluster with those AZGP1- BM cells, many OL cells cluster with the other AZGP1 cells, how do you explain this? Are there any interesting genes upregulated along the trajectory, potential transcription factors driving this transition?

We agree with the reviewer, that the analysis and interpretation of the AZGP1-neg. cluster needed to be revised. We therefore, repeated the analysis and added more aspects (see new Figure 6) – also in agreement with comments from reviewer 1. Trajectory analysis (Figure 6 B) was performed on the entire set of malignant plasma cells. Integrated analysis of the anchored datasets revealed that AZGP1 was – amongst others – significantly up-regulated in the majority of malignant plasma cells from the bone marrow and virtually not expressed in plasma cells from the lesion. Differential expression analysis showed that bone marrow cells from cluster 2 did not express AZGP1. To visualize top up-/downregulated genes of clusters along the trajectory, we have added a heatmap with the respective information (Figure 6B). Interestingly, also genes that were up-regulated in plasma cells from osteolytic lesions in other patients (like LAMP5 and HGF) were identified in cluster 2. To further investigate the biological background of the respective cells, we performed a gene set enrichment analysis and identified enriched EMT signatures being accompanied by down-regulation of pathways associated with proliferation (Figure 6C). These findings of EMT-like transformation of bone marrow plasma cells in a patient with para-medullary spread provide for the first time clinical evidence for pre-clinically developed models that established the role of EMT in MM (e.g. Roccaro et al, Cell Rep, 2016, Azab et al., Blood, 2012).

The new analyses are presented in Figure 6 and we changed the paragraph in the results section on page 12.

- There was no further experimental validation of the potentially interesting subpopulations (STMN1 TUBA1B subtype) or the DE genes between OL and BM lesions or changes upon therapy, for example in situ validation or by isolating those subpopulations. While it is appreciated that STMN1 and TUBA1B are associated with shorter survival, there were also no further functional analyses.

We agree with the reviewer and added the respective limitation of missing *in situ* validation in results and discussion.

In line with the comment from reviewer 3, we aimed at further investigating the possible prognostic implications of the STMN1-pos clusters. As mentioned in the first version of the manuscript, STMN1 is also included in the IFM GEP score, which consists mainly of cell cycle genes and genes associated with chromosomal instability. To objectify the prognostic relevance of the identified clusters, we calculated the UAMS17 score for every individual single cell and visualized expression together with STMN1 in UMAPs. We demonstrate that STMN1-pos clusters also show higher UAMS17 scores, which underlines the possible prognostic implications.

This is now included as panel J in the new figure 4.

We appreciate the comment from reviewer 2 and agree that a functional validation of our findings would have expanded the significance of our findings. However, the primary scope of our translational trial was to establish single cell RNA sequencing on clinical samples from different patients, locations and stages of MM to proof feasibility of our workflow.

As mentioned above, we included the limitation of *in situ* validation in the discussion.

Discussion, page 17:

Limitations of this study include the small number of analyzed patients and the lack of preclinical validation. Therefore, our findings regarding para-medullary spread of disease and residual MM after therapy are to be interpreted with caution.

Reviewer #3 (Remarks to the Author): Expert in multiple myeloma genomics and pathogenesis

Merz and colleagues describe scRNA-Seq from paired random bone marrow and osteolytic lesion sites in 10 patients with myeloma. They also describe longitudinal scRNA-Seq from 3 patients. The manuscript is beautifully written and very easy to follow. They do not waste the reader's time by extensively recapitulating findings from bulk RNA-Seq (though they do provide brief analyses that give us confidence in the data), but instead focus on examining the intra-patient comparisons which are enabled by this technology. Key findings are the differential expression of genes between the BM and OL sites within patients and the differential expression of genes between diagnosis samples and MRD samples. I believe that this paper adds significantly to the myeloma scRNA-Seq literature.

We thank the reviewer for the evaluation and appreciate the comment on bulk RNA-seq and our effort to provide a connection to scRNA-seq. It was exactly our intention to show as a proof of principle that the long history of classifications and prognostication from bulk RNA-seq can be transferred to scSeq data.

I have only minor comments:

The survival comparisons in fig. S2 are for allcomers. CoMMpass represents very heterogeneously treated patients and it may be that this is affecting the survival analysis. There should be sufficient patient numbers in CoMMpass to also look at broad treatment groups (say IMiD versus PI versus joint IMiD/PI therapy). It may be that expression of these genes is relevant for particular treatments.

We thank the reviewer for this interesting approach that led to further analyses. When looking at the CoMMpass datasets, we indeed found that e.g. patients treated only with bortezomib-based regimen during induction therapy and higher STMN1 expression levels experienced adverse outcome compared to patients treated with other regimen. The best outcome was observed in patient treated with PI/IMiD combinations and lower STMN1 expression levels (see below). This new figure is included now as supplemental figure 2C.

However, treatment selection itself might have introduced a selection bias into the survival analyses of the CoMMpass cohort. Younger/fit patients were most likely treated with RVD during induction compared to frail individuals. To objectify the prognostic significance of the STMN1-pos cluster we repeated and expanded the analyses also according to comments from reviewers 1 and 2. Beyond STMN1 and TUBA1B, we found overexpression of genes associated with proliferation in the respective clusters. Proliferation is a central prognostic marker in MM. Furthermore, the respective clusters showed higher values for the UAMS17 score, which identifies high-risk patients regardless of the applied treatment. This underlines the possible prognostic implications of the identified clusters.

	Progression-free survival		Overall survival	
	Hazard ratio	p-value	Hazard ratio	P-value
— STMN1 high and PI / IMiD combination	1.31	<0.0001	1.41	0.0003
— STMN1 high and bortezomib-based	1.45	<0.0001	1.47	<0.0001
— STMN1 low and PI / IMiD combination	0.64	<0.0001	0.63	<0.0001
— STMN1 low and bortezomib-based	1.14	0.0327	1.31	0.0046

Although I think I understand, it is not explained entirely clearly why, for example, DKK1 differs between OL and BM after anchoring (page 10) but not in the earlier analysis (page 6). Although the anchoring procedure is described in the reference and methods, a line or two of explanation within the results section would be helpful to understand the discrepancy.

At first, we merged both datasets without integration and performed the standard Seurat downstream analysis. Although the same plasma cell clusters were identified in the marrow and osteolytic lesion, the merging process alone did not allow the direct comparison of cells identified in both location - lesion and marrow.

Since we achieved high purity of our plasma cell populations after sorting from both locations, our ultimate goal was to identify the differences in single cell gene expression in very homogenous cell populations. Here, anchoring both datasets for integrated analysis made it possible to compare single cell gene expression between cells from same clusters identified under both conditions (e.g. in our case lesion versus marrow and before versus after treatment).

We have added a paragraph in materials and methods to explain the purpose of the integrated analysis:

Results, page 11:

To identify genes that are differentially expressed in PC from both conditions, we performed an integrated analysis after anchoring datasets from OL and BM for each individual patient as described before (Butler et al., Nature biotechnology, 2018). This process identifies pairwise correspondences between PC from different origins – called anchors – to transform datasets into a shared space. By aligning PC from BM and OL we were able to directly compare single PC gene expression from both locations. After applying the integration procedure, malignant PC were robustly detected in all datasets and the same PC clusters were identified in BM and OL.

In a similar vein, can we be reasonably confident that these differences between OL and BM would not have been discovered by bulk sequencing of samples from both sites followed by some sort of paired analysis (e.g. Wilcoxon signed rank test)? I.e. it would be good to fully justify the use of scRNA-Seq for these key differential expression findings.

The reviewer is absolutely right in that the final validation of our hypothesis would have been the direct comparison with bulk RNA-seq or GEP data. However, due to the limited number of cells, we were not able to perform scRNA-seq, WES and bulk RNA-seq or GEP on the same samples.

Given the small percentage of STMN1 pos. cells as shown now in figure 4G that contributed to the total number of sequenced plasma cells, it can be speculated that the respective signal would have likely been missed in bulk data.

With regards to the comparison between lesion and marrow, the same inference can be made. As indicated in figure 5E, the same plasma cell clusters were identified in marrow and lesion. The differences e.g. in DKK1 and HGF expression were very subtle and only detected by scRNA-seq analysis. Furthermore, as depicted in the

respective figure, the number of cells expressing the respective genes was different. Therefore, subtle differences in expression levels as well as number of cells between lesion and marrow were detected by the scRNA-seq analyses while past bulk GEP studies did not report on such findings.

Page 11: "AZGP1 is a known tumor suppressor gene causing mesenchymal-to-epithelial transition and inhibiting invasion and metastasis". Should this not read "AZGP1 is a known tumor suppressor gene whose downregulation causes mesenchymal-to-epithelial transition..." My understanding of the differential expression of AZGP1 is that it was seen in a single sample. If this is correct, the authors should caution the reader against over-interpretation of the AZGP1 data (although the pseudotime analysis is extremely nice!).

We thank the reviewer for this comment and corrected the respective passage.

Results, page 12:

AZGP1 is a known tumor suppressor gene and its loss causes epithelial-to-mesenchymal transition (EMT).

Furthermore, in line with comments from reviewer 1 and 2, we toned down the interpretation of our finding.

Results, page 14:

However, our observations were based on only three patients showing feasibility of this methodology to characterize residual cells. Future longitudinal studies in larger cohorts of homogeneously treated patients are needed to confirm these initial findings.

Discussion, page 17:

Limitations of this study include the small number of analyzed patients and the lack of preclinical validation. Therefore, our findings regarding para-medullary spread of disease and residual MM after therapy are to be interpreted with caution.

Reviewer #4 (Remarks to the Author): Expert in osteolytic lesion detection and haematologic malignancies

The authors reported comparative single-cell RNA sequencing analysis of bone marrow plasma cells from random bone marrow aspirate and image-guided biopsy of the osteolytic lesions in patients with newly diagnosed and relapsed myeloma. Clonal heterogeneity in the same patient has well been recognized in myeloma. In addition, the spatial heterogeneity between the focal lesions and traditional iliac crest biopsies has been documented in large study by the UAMS group using numerical or structural chromosomal aberrations, short insertions or deletions or single nucleotide variants. This study uses single cell RNA sequencing, a much more sensitive technology that allows a deeper dive the single-cell transcriptomes. The study uses small sample size of 7 newly diagnosed and 3 relapsed cases and therefore the results are thought-provoking and represent a feasibility study and a proof of concept. Clinical correlation is limited and require a larger study. Paired sample analysis of the residual tumor cells post-treatment is interesting but whether these cells represent a driver clone for relapse is not clear. In addition, whether the specific genes preferentially expressed in the OL PCs compared to BM PCs represent a subclone or expression alteration due to factors in the tumor microenvironment remain unknown, and perhaps should be discussed. The intra and inter- patient heterogeneity are as to be expected but intrigue data is in the much different single cell transcriptomics identified when WES only demonstrate limited differentially expressed genes.

Based on the objective of this report being the technical feasibility, more details should be provided on the chosen patients and the biopsies. Clinical information regarding the extent of bone involvement is needed. The number and size of the lesions were not described. One lytic lesion biopsy was obtained in each patient (except for one case where 2 sites were chosen) at the same time as the bone marrow aspirates. To fully understand how representative the one site may be of all other lytic lesions, such description would be needed.

We appreciate the comment by reviewer 4. In line with comment 3 by the same reviewer we expanded the information on how representative the biopsied lesions were and how the patients were selected within the study.

Methods, pages 18 and 19:

Every patient with new OL on PET/CT was discussed in the Roswell Park Multiple Myeloma multi-disciplinary tumor board. If a patient had multiple new OL, the interventional radiologists (AB and RA) determined if a lesion was accessible for an imaging-guided biopsy with the least risk for peri-interventional complications. Lesions had to be identified on CT, showing bone destruction. OL PET-positivity was not required since some OL may not be PET-avid at primary diagnosis and false-negative PET scans can occur based on low hexokinase-2 expression.

It is important to mention that with the exception of patient RRMM01, all lesions were strictly intra-medullary and therefore representative for the remaining numerous lesions in each patient. This is important since the first study on spatial genomic

heterogeneity in MM found a connection between lesion size and unshared mutations. In the materials and methods section of the respective paper that was published in Nature communications in 2017, authors described that lesions were identified by either MRI in 32 of 42 patients if lesions were larger than 1 cm in diameter or identified by PET/CT. In our study, lesions were primarily identified by whole body CT that has a higher resolution to detect osteolytic lesions. E.g. Figure 1 shows now one of the largest lesions that was biopsied within our study that measured 1.5 cm in CT. Therefore, with the exception of RRMM01 with para-medullary disease, the remaining lesions would have been assigned to the smallest categories in the study by Rasche et al. In agreement with their study, we found limited evidence for spatial heterogeneity in these small, strictly intra-medullary lesions. This is an important point and is now included in the discussion.

Results, page 6:

With the exception of patient RRMM01 with para-medullary spread from an OL of the clavicle, all samples were acquired from strictly intra-medullary lesions.

Discussion, page 15 and 16:

Possible explanations for these differences might be the retrospective nature (analyzing previously collected and stored samples) of the Rasche et al study and the inclusion of more patients with advanced disease (larger lesions and extramedullary disease). This may have introduced sampling bias as there was a strong connection between spatial heterogeneity and lesion size. In the current study using prospective acquisition of fresh PC from intra-medullary locations, we demonstrated that there were greater than 80% shared mutations between both locations with the exception of patient RRMM01 with para-medullary disease and significant spatial heterogeneity. This is in agreement with the study by Rasche et al. that found limited spatial heterogeneity from WES in patients with smaller lesions.

We thank the reviewer for this comment, since it also let to a new comparison with the only existing data on spatial heterogeneity in MM from Rasche et al.

The technical aspect of this study is of interest as their established method may inform other investigators. The authors should be commended for the concurrent biopsies and processing, to avoid the interference of processing artifact. While sample size determination for scRNAseq continues to be a controversial topic and standardized methodology has not been determined, the number of PC examined in this study is in line with others that have used the similar technique for study of drug resistance mechanism or progression from pre-myeloma conditions. Please further specify whether there were cases that have been excluded due to inadequate PCs.

Determining feasibility of our approach was an important first step in our translational clinical study and we appreciate the reviewer's comment. To illustrate our process, we added the new figure 1 also in accordance with comments from reviewer 1. No cases had to be excluded due to inadequate number, viability or purity of plasma cells or due to not passing *in silico* QC after scRNA-seq. To be transparent about our sample quality, we provide information on bone marrow infiltration rate, QC assessment with flow after magnetic bead sorting as well as QC parameters after scRNA-seq in table 1. The comparison of cells from osteolytic lesion and bone

marrow did not reveal any significant differences regarding these parameters underlining the feasibility of our workflow.

Since PET/CT was used to identify these lesions, please also consider adding the information regarding FDG uptake of the chosen lesion. The decision for which lesion to chose was done by the multidisciplinary tumor board. Please discuss the general factors included in the decision on the biopsy sites.

This is an important In line with comment 1, we provide more information on the selection process in materials and methods as described above in the answer to comment 1. We also clarify interpretation and relevance of PET in our study.

Methods, page 18:

Interpretation was performed according to the Interpretation criteria for FDG PET/CT in multiple myeloma (IMPeTUs). OL were characterized by the presence of circumscribed areas with bone loss. Increased tracer uptake was graded according to the 5-point Deauville scoring system.

Methods, page 18 and 19:

Lesions had to be identified on CT, showing bone destruction. OL PET-positivity was not required since some OL may not be PET-avid at primary diagnosis and false-negative PET scans can occur based on low hexokinase-2 expression.

PC viability from both biopsies was excellent and all cases had enough number of PCs that passed QC. Aspiration of lytic lesions is typically done by fine needle aspiration using a small-gauge needle compared to the bone marrow aspiration needles. However, the percent PC in the focal lesion is typically high, usually represent clusters of PCs. Was there an immediate pathologic evaluation by cytopathology to assure an adequate number of cells were obtained?

The reviewer raises a very important point that we had to consider when planning our prospective collection of samples within a translational study. In a previous study, we had already confirmed – as stated correctly by the reviewer – that plasma cell infiltration rate in osteolytic lesions is higher than in corresponding regular bone marrow aspirates (https://ascopubs.org/doi/10.1200/JCO.2016.34.15_suppl.8040). Therefore, we did not implement immediate, on-site evaluation by cytopathology since we assumed that the number of plasma cells from the lesion would actually surpass the infiltration rate of the random bone marrow aspirate. However, since both aspirates – from bone marrow and lesion – were collected in the same session, we were able to assess quality of both samples on the same day before sequencing. To illustrate this process that ensured high plasma cell purity and viability in both locations, we modified the respective figure 1 as shown in the respective comment to reviewer 1. The immediate handling of samples on the same day of the procedure included automated cell counting and viability assessment after CD138-sorting and flow cytometric quantification of plasma cell concentration. All cells passed this first QC and were transferred to scRNA-seq.

The usual standard-operating-procedure within our trial was: After discussing the patient in the tumor board, informed consent and scheduling of the procedure, the

patient arrived at 7:30 am in interventional radiology at the day of the intervention. By 08:00 am interventional radiologist (AB and RA) started the procedure with laboratory staff on-site (MM). Right after both samples are acquired (09:00 – 10:00 am), they were transferred to the research lab for plasma cell separation (until 11:00 – 12:00 am). Results from flow cytometry (JT, PW) for QC were available around 12:00 am – 01:00 pm and samples were transferred to scRNA-seq (MM, JL, SG, PS). This workflow ensured that results on how many cells were captured in the procedure were present within few hours. We first thought about including a step with immediate evaluation by cytopathology, but decided against this step based on our previous results of higher plasma cell content in lesions.

Overall, the work is innovative and now further highlights the inter-lesion heterogeneity and incorporation of concurrent biopsy from multiple sites pre- and post-treatment for targeted and personalized therapy in this incurable disease.

We thank the reviewer for this positive statement.

REVIEWER COMMENTS

Reviewer #1 (Remarks to the Author):

The authors have addressed my comments.

Reviewer #2 (Remarks to the Author):

The authors have addressed all of my comments. But I have a remark on the comment 10 from Reviewer 1. Looking also at the heatmap of figure 6B, I can see that erythrocytes markers such as HBB, HBA1, HBD, HBM come up in the list of top DE genes. Did the authors check for doublets using also a computational tool or manually looking at canonical marker genes (they mention that they removed multiplets by removing cells with high number of detected genes, but this procedure does not remove all doublets). If there are no doublets, another potential option could be ambient RNA which is very common when there are erythrocytes in a single-cell dataset (the authors said that they detected and removed 388 erythrocytes). If this is the issue, then tools such as DecontX should be able to resolve this.

Reviewer #3 (Remarks to the Author):

The authors have fully addressed all my comments about the manuscript.

Reviewer #4 (Remarks to the Author):

The revised manuscript in response to prior comments have much improve the clarity of the manuscript.

My prior comments focused on the technicality of OL selection and sample processing, before analysis since this basic step would critically impact the quality of the samples for analyses, and therefore the data quality. To this, the authors have adequately provided more detailed workflow.

My additional comments are minor and relate to pilot nature of the study, which is thought provoking, but perhaps overstretched in the discussion and conclusion. The authors state the study limitation specifically the small sample size and lack of prospective clinical and in vitro validation. With that in mind, their description of the importance or link of the specific gene changes they found and clinical phenotype should be toned down to reflect just hypothesis, not a causal link.

1. "We provide for first time clinical evidence for the pre-clinical hypothesis that hypoxia-driven EMT-like processes drive extramedullary spread of MM (30). Our findings and recent preclinical studies (28) support the hypothesis that OL are derived from a common PC ancestor developing molecular features to cause myeloma bone disease. Thus, MM would behave like a solid tumor with PC metastasizing to distant locations, inducing OL."

In theory, all lytic lesions are metastasis from the original site where from is unknown when patients present with myeloma, a systemic disease. Reference 28 and 30 referred to animal models where tumor cells originate at one place (in this case, tumor colonized bones in SCID-rab model more like isolate plasmacytoma), with subsequent detection of some tumor clones in distant bone sites. Since AZGP1 downregulation in OL PCs occurs in this para-medullary case but not in other "metastatic" lytic lesion, it could be a stretch to say this is an evidence of the EMT transition and extramedullary spreading during myeloma progression. Para-medullary disease is a soft tissue component extending from intramedullary cavity typically through cortex erosion. They are more similar clinically to lytic lesions than a classic extramedullary spread such as soft tissue or visceral plasmacytoma, which are more like metastatic solid tumors such as pancreas cancer, in their reference.

2. post therapy PC profiling: "These results indicate that scRNA-seq is able to identify and characterize residual disease. The respective changes in single PC

transcriptomes may help identify new strategies to eradicate MRD in the future”

Data on contribution of these genes in drug resistance has been mostly in vitro. Further comparative studies of these gene expression in baseline and post treatment to validate their roles in drug resistance, as well as the authors’ own analysis of the PCs in this case at the time of relapse will clarify its significance and open a new therapeutic venue. At this time, it is unclear whether these genetic changes are important for survival of the MRD clone noted and whether they contribute to relapse, based on only 2 cases. The PR case post RVD-Daratumumab is very interesting as this is one of the most potent anti-myeloma regimens currently utilized. The authors present data of all 3 cases together, but is there anything unique about this case? If so, that would be of great interest to the readers.

3. Since there are both newly diagnosed cases and RRMM, are the gene changes in the post treatment MRD marrow and the PR case post RVD-dara more similar to the RRMM? That would give more weight to the importance of these genes in relapse and drug resistance

Reviewer #2 (Remarks to the Author):

The authors have addressed all of my comments. But I have a remark on the comment 10 from Reviewer 1. Looking also at the heatmap of figure 6B, I can see that erythrocytes markers such as HBB, HBA1, HBD, HBM come up in the list of top DE genes. Did the authors check for doublets using also a computational tool or manually looking at canonical marker genes (they mention that they removed multiplets by removing cells with high number of detected genes, but this procedure does not remove all doublets). If there are no doublets, another potential option could be ambient RNA which is very common when there are erythrocytes in a single-cell dataset (the authors said that they detected and removed 388 erythrocytes). If this is the issue, then tools such as DecontX should be able to resolve this.

We thank the reviewer for the opportunity to clarify why the respective hemoglobin genes are listed in the heatmap of figure 6B:

The heatmap was derived from the differential expression analysis of the entire dataset of patient RRMM01, including plasma cells and non-malignant, non-plasma cells. For better legibility, we filtered out non-plasma cells and showed only results for plasma cell clusters in figure 6B of the revised manuscript. Therefore, hemoglobin genes are still present in the respective heatmap but are not expressed by plasma cells, as indicated by black bars. This lack of homogeneous expression of hemoglobin genes in plasma cells becomes more evident when including the non-plasma cell clusters in the heatmap as shown on the next page (page 2 of this document). It can be appreciated that cluster 9 from the entire dataset (**A**) was populated by cells expressing the respective markers of erythroid lineage as demonstrated by the bright yellow bars in the heatmap (**B**). Cell type annotation with SingleR confirmed that the respective cluster represented erythrocytes (**C**). We would prefer to keep the respective genes listed in the heatmap since it represents the unfiltered results of the differential expression analysis. We include the information that the heatmap shows only results for plasma cells now in the respective figure legend.

Although we are convinced that this addresses the issue raised by reviewer 2, we were intrigued by the comment regarding multiplets and especially contamination with ambient RNA. Therefore we revisited the raw data regarding number of detected features and applied DecontX to our dataset.

On the next page (page 4 of this document), we show plasma cell clusters detected in the entire dataset (bone marrow and osteolytic lesion samples (**A**)). As expected, there was a strong correlation between number of detected genes and detected RNA molecules with only few outliers (**B**). Violinplots of detected features in the respective clusters showed that a more conservative approach (e.g. filtering out cells with more than 6000, 5000 or 4000 detected features) would not have affected the majority of identified plasma cells (**C**).

Although we implemented a thorough pre-analytical and *in silico* filtering process of low-quality cells and non-malignant, non-plasma cells as described in figure 1 of the revised manuscript, we became very interested in checking our dataset for ambient RNA as proposed by reviewer 2. For this purpose, we converted the Seurat object containing data from the entire experiment into a SingleCellExperiment object to run DecontX. Results are summarized on page 5 of this document. First, we investigated whether cells clustered together based on the origin of the samples (**A**). In line with our analyses in Seurat, cells from individual patients and not locations clustered together. Panel (**B**) on page 5 shows the color-coded UMAP of results from DecontX, ranging from no contamination in blue to 100% contamination with ambient RNA in red. The only cluster that were significantly affected by contamination with ambient RNA was cluster 6 and to a lesser extend cluster 5 (**C**). Both clusters were populated by non-malignant cells from multiple different patients as shown in panel (**A**). Importantly, the remaining clusters that were clearly formed by cells from individual patients (bone marrow and osteolytic lesions) and were used for the further analyses showed no significant contamination (**B**). This underlines that the observed results were not caused by ambient RNA.

Taken together, we thank the reviewer for the opportunity to clarify, why hemoglobin genes are listed in the heatmap in Figure 6B. Furthermore, we thank the reviewer for the comment regarding ambient RNA. This will be very helpful for future single cell experiments and will become a standard in our downstream analyses.

A Clusters identified in entire dataset (marrows + lesions)

B Scatterplot of detected features and RNA counts

C Violinplot of detected RNA features by cluster

Reviewer #4 (Remarks to the Author):

The revised manuscript in response to prior comments have much improve the clarity of the manuscript. My prior comments focused on the technicality of OL selection and sample processing, before analysis since this basic step would critically impact the quality of the samples for analyses, and therefore the data quality. To this, the authors have adequately provided more detailed workflow.

My additional comments are minor and relate to pilot nature of the study, which is thought provoking, but perhaps overstretched in the discussion and conclusion. The authors state the study limitation specifically the small sample size and lack of prospective clinical and in vitro validation. With that in mind, their description of the importance or link of the specific gene changes they found and clinical phenotype should be toned down to reflect just hypothesis, not a causal link.

1. "We provide for first time clinical evidence for the pre-clinical hypothesis that hypoxia-driven EMT-like processes drive extramedullary spread of MM (30). Our findings and recent preclinical studies (28) support the hypothesis that OL are derived from a common PC ancestor developing molecular features to cause myeloma bone disease. Thus, MM would behave like a solid tumor with PC metastasizing to distant locations, inducing OL."

In theory, all lytic lesions are metastasis from the original site where from is unknown when patients present with myeloma, a systemic disease. Reference 28 and 30 referred to animal models where tumor cells originate at one place (in this case, tumor colonized bones in SCID-rab model more like isolate plasmacytoma), with subsequent detection of some tumor clones in distant bone sites. Since AZGP1 downregulation in OL PCs occurs in this para-medullary case but not in other "metastatic" lytic lesion, it could be a stretch to say this is an evidence of the EMT transition and extramedullary spreading during myeloma progression. Para-medullary disease is a soft tissue component extending from intramedullary cavity typically through cortex erosion. They are mire similar clinically to lytic lesions than a classic extramedullary spread such as soft tissue or visceral plasmacytoma, which are more like metastatic solid tumors such as pancreas cancer, in their reference.

We agree with the reviewer that downregulation of AZGP1 alone does not explain extra-/para-medullary spread of the disease and it needs to be mentioned that the pathogenesis of para- and extra-medullary myeloma might be different from each other. The latter one will be subject to future analyses. We therefore include the following sentence in the discussion.

Discussion, Page 17:

Establishing a translational workflow for scRNA-seq from clinical samples from different locations was one of the major goals of our study. Based on our results, future analyses will include larger numbers of patients and also patients with extramedullary disease in addition to para-medullary MM, to decipher the biological differences between both conditions.

2. post therapy PC profiling: “These results indicate that scRNA-seq is able to identify and characterize residual disease. The respective changes in single PC transcriptomes may help identify new strategies to eradicate MRD in the future” Data on contribution of these genes in drug resistance has been mostly in vitro. Further comparative studies of these gene expression in baseline and post treatment to validate their roles in drug resistance, as well as the authors’ own analysis of the PCs in this case at the time of relapse will clarify its significance and open a new therapeutic venue. At this time, it is unclear whether these genetic changes are important for survival of the MRD clone noted and whether they contribute to relapse, based on only 2 cases. The PR case post RVD-Daratumumab is very interesting as this is one of the most potent anti-myeloma regimens currently utilized. The authors present data of all 3 cases together, but is there anything unique about this case? If so, that would be of great interest to the readers.

This is a very interesting point that we will investigate in the future. Based on our analyses using non-negative matrix factorization of the entire pre-/post samples, we were able to identify genes that are up- and down-regulated upon treatment in all three patients. We choose this option, since all three patients received RVD during induction. In agreement with this hypothesis, we found differentially expressed genes that have been associated with steroid exposure or proteasome inhibition. We agree, that deciphering plasma cell response to daratumumab or any other CD38-antibody in residual cells would be of great interest, since quadruplet regimen are the most effective treatment options during first line at the moment and early resistance to the respective agents is associated with dismal outcome. Future analyses will focus on pre-/post anti-CD38 treated patients to address this very interesting

point raised by the reviewer. This limitation and outlook is mentioned in the discussion on page 17.

3. Since there are both newly diagnosed cases and RRMM, are the gene changes in the post treatment MRD marrow and the PR case post RVD-dara more similar to the RRMM? That would give more weight to the importance of these genes in relapse and drug resistance

This is a very intriguing idea that we also investigated when we first analyzed our dataset and compared RRMM with NDMM and post-treatment samples. However, as described on pages 6 and 7 of the manuscript, inter-patient heterogeneity was the major determinant between plasma cells. That means that heterogeneity introduced by patient specific genetic events like a hyperdiploid karyotype or IgH translocation outweighed factors like location (bone marrow versus osteolytic lesion) or treatment situation (NDMM versus RRMM versus NDMM after therapy). Nevertheless, as described in answer 2 to this reviewers comments, we were able to identify changes spanning all three patients with NDMM. These changes were consistent with exposure to the respective drugs while not comparable to general transcriptional changes in NDMM or RRMM patients. An explanation for this finding might be, that the post-treatment samples were collected rather early in the patients' cause of the disease (after induction). This means that from a clinical point of view, they did not fulfill the criterion of relapsed or refractory MM with e.g. rising M-protein after last line of therapy. In fact, the respective patients were in serological remission but showed evidence for residual plasma cells. Therefore, we think that MRD is different from relapsed or refractory disease, which is also underlined by the different prognosis of these two patient groups: While an MRD-positive patient after first line treatment can live up to years without indication for new treatment, patients with RRMM usually show dismal survival without timely therapeutic interventions. Based on our results, scRNA-seq might help to identify strategies to eradicate MRD in the future, which is mentioned in the discussion on pages 16-17.

We thank the reviewer for this thought-provoking impulse. In this particular case (MRD-positivity versus relapsed/refractory disease), myeloma behaves differently compared to other hematological disease like AML, where MRD-positivity triggers treatment decisions like allogeneic transplantation.

REVIEWERS' COMMENT

Reviewer #2 (Remarks to the Author):

The authors addressed all of my comments.

Reviewer #4 (Remarks to the Author):

The authors have addressed all of my comments. The quality of the manuscript has much improved. I do not have additional concerns